# Trends in snakebite deaths in India from 2000 to 2019 in a nationally representative mortality study

Wilson Suraweera[1]*, David Warrell[2]*, Romulus Whitaker[3]*, Geetha Menon[4]*, Rashmi Rodrigues[5]*, Sze Hang Fu[1]*, Rehana Begum[1], Prabha Sati[1], Kapila Piyasena[1]*, Mehak Bhatia[1], Patrick Brown[1,6]*, Prabhat Jha[1]*

[1]Centre for Global Health Research, Unity Health Toronto, and Dalla Lana School of Public Health, University of Toronto, Ontario, Canada; [2]Nuffield Department of Clinical Medicine, University of Oxford, Oxford, United Kingdom; [3]Centre for Herpetology/Madras Crocodile Bank, Vadanemmeli Village, Chennai, India; [4]Indian Council of Medical Research, Ansari Nagar, New Delhi, India; [5]Department of Community Health, St. John's Medical College, St. John's National Academy of Health Sciences, Bangalore, India; [6]Department of Statistical Sciences, University of Toronto, Toronto, Canada

**\*For correspondence:**
Wilson.Suraweera@unityhealth.to (WS);
david.warrell@ndm.ox.ac.uk (DW);
kingcobra@gmail.com (RW);
menongr.hq@icmr.gov.in (GM);
rashmijr@gmail.com (RR);
Hana.Fu@unityhealth.to (SHF);
delpepiyasena@gmail.com (KP);
patrick.brown@utoronto.ca (PB);
Prabhat.jha@utoronto.ca (PJ)

**Abstract** The World Health Organization call to halve global snakebite deaths by 2030 will require substantial progress in India. We analyzed 2833 snakebite deaths from 611,483 verbal autopsies in the nationally representative Indian Million Death Study from 2001 to 2014, and conducted a systematic literature review from 2000 to 2019 covering 87,590 snakebites. We estimate that India had 1.2 million snakebite deaths (average 58,000/year) from 2000 to 2019. Nearly half occurred at ages 30–69 years and over a quarter in children < 15 years. Most occurred at home in the rural areas. About 70% occurred in eight higher burden states and half during the rainy season and at low altitude. The risk of an Indian dying from snakebite before age 70 is about 1 in 250, but notably higher in some areas. More crudely, we estimate 1.11–1.77 million bites in 2015, of which 70% showed symptoms of envenomation. Prevention and treatment strategies might substantially reduce snakebite mortality in India.

## Introduction

The World Health Organization (WHO) estimates that 81,000–138,000 people die each year from snakebites worldwide, and about three times that number survive and but are left with amputations and permanent disabilities (*World Health Organization (WHO), 2019a*). Bites by venomous snakes can cause acute medical emergencies involving shock, paralysis, hemorrhage, acute kidney injury and severe local tissue destruction that can prove fatal or lead to permanent disability if left untreated. Most deaths and serious consequences from snakebite envenomation (exposure to venom toxins from the bite) are avoidable by timely access to safe and effective antivenoms (*Gutiérrez et al., 2017*). Snakebite deaths and envenomation are largely neglected topics in global health. However, in 2017, the WHO included snakebite envenoming in the priority list of neglected tropical diseases (*World Health Organization (WHO), 2019b*) and launched in 2019 a strategy for prevention and control of snakebite, aiming to halve the numbers of deaths and cases of serious disability by 2030 as compared to 2015 baseline (*World Health Organization (WHO), 2019c*). Achieving this goal will require substantial progress in India, which is home to approximately half of global snakebite deaths. Snakebite deaths and disability remain a major public health challenge also for poor rural communities in many parts of Asia, Africa, Latin America and Oceania.

Direct estimation of 46,000 annual snakebite deaths in India in 2005 (*Mohapatra et al., 2011*) prompted a revision of the WHO's global total, which had estimated about that number for the entire world. The 2005 Indian estimate relied upon analyses of about 123,000 verbal autopsy records from 2001 to 2003 in the Registrar General of India's (RGI) Million Death Study (MDS), one of the largest nationally representative mortality surveys. Now the MDS has reported cause-specific mortality patterns on over 600,000 deaths from 2001 to 2014 for the whole of India. Here, we report seasonal and temporal trends in snakebite mortality over the last two decades in India and its spatial distribution. We provide estimates of total snakebite deaths for the 20-year period 2000–2019 by age and sex. Our earlier report estimated a crude ratio of about one death to 20 envenomations. We now further quantify the levels of envenomations based on a systematic review of 88,000 snakebites in the published literature. The literature also provides details on the specific causes, bite locations, and treatment of envenomations. Finally, enhanced surveillance including facility-based tracking will be central to the Government of India's strategies to reduce snakebite deaths. Thus, we provide estimates on the degree to which snakebites and deaths are reported adequately in public facilities. *Appendix 1—figure 1* shows the overall study design, data sources, input resources and outcomes.

## Results

### Trends in snakebite mortality and its geographic and temporal patterns

From 2001 to 2014, the MDS reported deaths with causes classified by physicians who examined verbal autopsy records collected from over 3.6 million households in three distinct nationally representative sampling frames (1993–2003; 2004–13; and 2014–23). Two of 404 independent physicians coded each death to the International Classification of Diseases-10th revision (ICD-10), reconciling (anonymously) any coding differences with a senior physician adjudicating any persistent disagreements (*Gomes et al., 2017*; *Aleksandrowicz et al., 2014*; *Menon et al., 2019*). Among 611,483 available records, 2833 deaths were assigned to snakebites (ICD-10 code X20). The two physicians agreed on the diagnosis 92% of the time. About 94% of snakebite deaths occurred in rural areas, and 77% occurred out of hospital (*Appendix 1—table 1*).

We applied the age- and sex-specific proportion of snakebite deaths to total deaths as estimated by the United Nations Population Division (UN) for India (*United Nations, 2019*) to estimate national death rates by age and sex, as well as absolute totals for each year (*Table 1*). The UN totals are based on careful demographic review of census and other data sources. The fieldwork procedures of the Sample Registration System (SRS the underlying demographic survey on which the MDS is based) leads to some undercounts (of about 5–10%) of expected deaths (*Gerland, 2014*). The SRS is representative at the state and rural/urban strata, and has a large, distributed sampling covering over 7000 small areas in the whole of the country (*Registrar General of India, 2017*). Hence, any missing deaths are generally randomly distributed across states, and not clustered in one state or one key sub-group, such as in rural areas (*Dhingra et al., 2010*; *Aleksandrowicz et al., 2014*; *Menon et al., 2019*). Thus, the proportion of snakebite deaths is not likely an underestimate. However, total snakebite deaths might be underestimated. The use of the UN death totals adjusts for these possible undercounts and provides a plausible national total for each year.

Total snakebite deaths in India from 2001 to 2014 totaled about 808,000, with reasonably narrow uncertainty range of 738,000 to 833,000, based on both physicians immediately assigning snakebites or only one physician doing so. Some age-specific death rates fell, but as population growth averaged 1.1% annually, the application of annual age-specific rates to the UN death totals for that year showed that the overall number of snakebite deaths grew from about 55,000 in 2001 to about 61,000 in 2014. During the 2001–2014 MDS study period, the average age-standardized snakebite death rate (using the Indian census population of 2001 to take into account the minor change in age structure) was 4.8 per 100,000 population, falling annually by 0.8%.

Declines in the age-specific snakebite death rate were fastest for children aged 0–14 years (declining by about 1.6% annually), with slower declines in young adults aged 15–29 years (1.2% annually) and no declines among middle-aged adults (30–69 years). Before 2010, snakebite death rates were higher in boys than girls but from 2010 to 2014, death rates in girls exceeded those for boys (*Appendix 1—table 1*). The age-specific risks translate to a probability of 0.37% (uncertainty

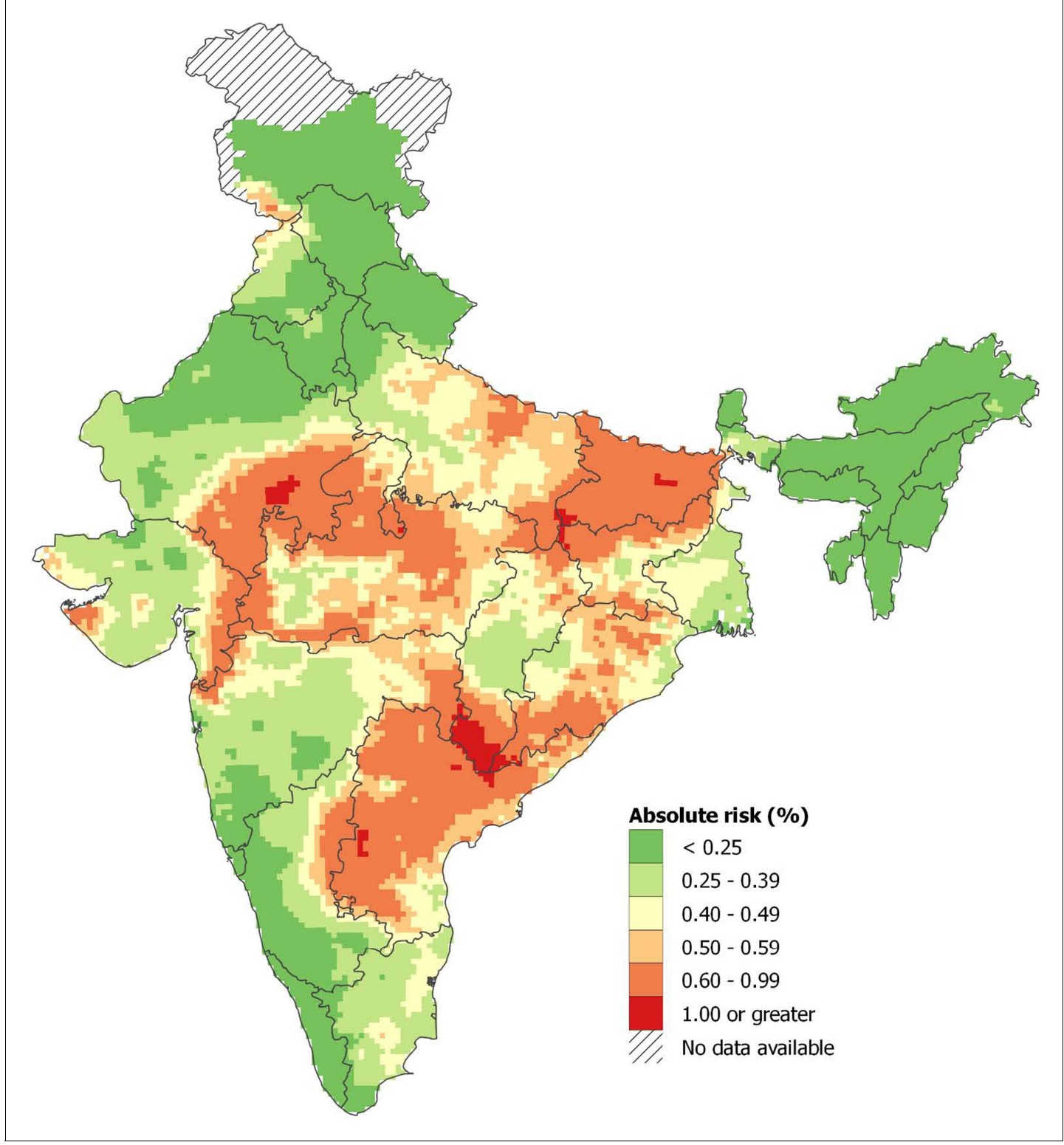

**Figure 1.** Spatial distribution of snakebite mortality risk in India for 2004-13. Note: About 0.33% of the Indian population lived in areas with an absolute risk of 1% or greater of dying from snakebite before age 70 years, and 21% lived in areas with absolute risk of 0.6% or higher. Population estimates used the Gridded Population of the World version 4 for year 2015 (*Center for International Earth Science Information Network - CIESIN - Columbia University, 2015*). Further details of statistical method and stochastic uncertainties of spatial mortality risk pertaining to these estimates are explained in Appendix 3.

**Table 1.** Snakebite deaths in the Million Death Study, age-standardized and age-specific mortality rates and risks in India from 2001-2014.

| Year | Study deaths from snakebite/all causes | Standardized death rate /100,000 (all ages) and age-specific rates / 100,000* | | | | Snakebite mortality risk[†] | Estimated national deaths (000)[‡] |
|---|---|---|---|---|---|---|---|
| | | All ages | 0-14 | 15-29 | 30-69 | | |
| 2001 | 199 /41826 | 5.3 | 5.4 | 3.6 | 5.9 | 0.40% | 55.0 |
| 2002 | 183 /41740 | 5.2 | 5.2 | 3.5 | 5.8 | 0.39% | 55.3 |
| 2003 | 179 /38798 | 5.1 | 5.0 | 3.4 | 5.8 | 0.38% | 55.8 |
| 2004 | 190 /37380 | 5.0 | 4.6 | 3.5 | 5.7 | 0.38% | 55.6 |
| 2005 | 244 /46755 | 4.9 | 4.8 | 3.4 | 6.4 | 0.40% | 60.8 |
| 2006 | 214 /47471 | 5.3 | 4.7 | 3.2 | 6.7 | 0.40% | 62.7 |
| 2007 | 225 /48536 | 5.3 | 4.5 | 3.0 | 6.4 | 0.39% | 61.0 |
| 2008 | 215 /47673 | 5.1 | 4.2 | 2.8 | 5.9 | 0.36% | 57.4 |
| 2009 | 183 /47873 | 4.7 | 3.9 | 2.6 | 5.3 | 0.33% | 53.8 |
| 2010 | 200 /45719 | 4.3 | 3.9 | 2.6 | 5.0 | 0.32% | 52.4 |
| 2011 | 185 /46099 | 4.2 | 4.0 | 2.7 | 5.1 | 0.33% | 54.9 |
| 2012 | 227 /46635 | 4.3 | 4.3 | 2.8 | 5.4 | 0.36% | 59.2 |
| 2013 | 214 /45331 | 4.6 | 4.4 | 3.0 | 5.8 | 0.38% | 62.3 |
| 2014 | 175 /29647 | 4.7 | 4.2 | 3.0 | 5.9 | 0.37% | 61.2 |
| 2001-2014 | 2833 /611483 | 4.8 | 4.5 | 3.1 | 5.8 | 0.37% | 807.5 |
| Plausible range (Lower, Upper)[§] | | (4.4, 5.0) | (4.1, 4.7) | (2.8, 3.2) | (5.3, 6.0) | (0.34%, 0.38%) | (738.2, 833.4) |

* Death rates were standardized to the Indian population in census year 2001 to take into account minor changes in the age distribution over time.

† The probability of dying due to snakebite before reaching age 70 years in the hypothetical absence of other competing causes of death. This was calculated by summing the 5-yearly standardized death rates from ages 0 to 69 years.

‡ Total death estimates at all ages were calculated by applying the MDS sample weighted proportion of deaths from snakebites, using weighted 3-yearly moving average, to the United Nations Population Division death totals.

§ Plausible ranges: The inherent variation in these estimates is not from the underlying demographic estimates but in the determination of primary causes of death. Therefore, we used plausible ranges based on independent cause assignment by two physicians and subsequent agreement on ICD-10 codes (X20 or X29). The lower bound was based on immediate agreement of both physicians and upper bound based on either of two physicians coding snakebite deaths.

range 0.34–0.38%) of dying from snakebite before age 70 years in the absence of competing mortality (*Table 1*). This suggests that the average risk of an Indian dying from snakebite prematurely before age 70 is approximately 1 in 250.

Because the risk of dying from snakebites has been stable from 2001 to 2014, we can make reasonably reliable forward projections from 2015 to 2019 and backward projections from 2001 to 2000 (*Table 2*). This reveals that 1.2 million snakebite deaths occurred over this 20-year period. Of these deaths, 602,000 occurred among males and 565,000 occurred among females. With both sexes combined, about 543,000 (47%) occurred in middle-age (30–69 years), 325,000 (28%) among children below 15 years, 197,000 (17%) among adults aged 15–29 years, and 102,000 (9%) among those over age 70 years. Using the agreement of one or two physicians on the cause yielded generally narrow uncertainty estimates for each sex and age groups.

From 2001 to 2014, just under 70% of these snakebite deaths occurred in eight states with about 55% of the population: Bihar, Jharkhand, Madhya Pradesh, Odisha, Uttar Pradesh, Andhra Pradesh (which includes Telangana, a recently defined state), Rajasthan and Gujarat (*Table 3*). In these high-burden states, the age-standardized death rate was about six per 100,000. Snakebite death rates generally rose over time in most high-burden states, particularly in Bihar, but fell in Andhra Pradesh. The remaining lower-burden states began the study period with age-standardized death rates of

**Table 2.** Estimated snakebite deaths in thousands by age and sex from 2000 to 2019 in India.

| Age range | Male (LL, UL) | Female (LL, UL) | Both (LL, UL) |
|---|---|---|---|
| 0-14 years | 149 (134, 154) | 176 (160, 180) | 325 (294, 334) |
| 15-29 years | 109 (102, 111) | 88 (82, 89) | 197 (184, 199) |
| 30-69 years | 290 (269, 303) | 253 (232, 260) | 543 (501, 564) |
| 70 years or above | 54 (45, 60) | 48 (44, 50) | 102 (89, 110) |
| All Ages | 602 (551, 626) | 565 (518, 578) | 1,167 (1068, 1204) |

Total deaths for 2001-2014 MDS study period were 807,500 (**Table 1**). Deaths for 2000-2019 were calculated by extrapolating these annual deaths. The extrapolated annual deaths in thousands for outside the study period were 54.0 for 2000, 62.3 for 2015, 62.0 for 2016, 61.4 for 2017, 60.3 for 2018 and 59.8 for 2019.

Lower limit (LL) and Upper limit (UL) are lower and upper uncertainty bounds for estimates. The major uncertainty in our analyses, however, is not the demographic totals, but the cause of death classification. Hence, the lower bound was based on immediate agreement of both physicians on the ICD-10 code for snakebite and upper bound based on either of two physicians coding as snakebite death.

about 3.7, which fell over time. *Figure 1* shows the absolute risk of dying from snakebite using data from 7400 small areas (the small sampling units used in the RGI's Sample Registration System for the MDS) from 2004 to 2013. The absolute risks were calculated applying spatially smoothed predictive relative risks from a spatial Poisson model to the overall national risk before age 70 years (of about 0.4%, *Table 1*) after adjusting for any differences in rural/urban status, female illiteracy levels, temperature, and altitude of local areas. We observed greater than 0.6% (1 in 167) mortality risk before age 70 years in the highest risk sub-areas of Andhra Pradesh, Odisha, Bihar, Uttar Pradesh, Madhya Pradesh, Chhattisgarh, and Rajasthan. About 260 million people lived in these areas in 2015, including about 4 million people living in hot spots that had a 1% or greater risk of death from snakebite. Appendix 3 provides statistical details and credible intervals of these risk estimates.

Half of all snakebite deaths occurred during the southwest monsoon seasons from June to September. Seasonality was similar in each of the study years (*Appendix 1—figure 2*) and was similar in higher-burden and lower-burden states (data not shown). We used a Poisson time series model for snakebite deaths from 2001 to 2014 to predict the average daily snakebite mortality in India. The peak (294 deaths per day) was in mid-July and the trough (78 deaths per day) was in mid-February (*Figure 2A*). We also aggregated the deaths from 2001 to 2013 by every 100 m of altitude above sea level (*Figure 2B*). The crude death rates in areas below 400 m were about three times those in areas at about 1000 m. Over 80% of snakebite deaths occurred below 400 m and 50% occurred below 200 m.

## Characteristics of snakebites from a systematic review of the literature

A systematic literature review yielded 87,590 snakebite cases in India (both fatal and non-fatal) from 2000 to 2019 based on screening 1417 papers and including 78 studies from 24 states or union territories in India (*Figure 3*, *Appendix 2—figure 1*, *Appendix 2—table 2*). In the published studies, snakebites were more common in males (59%), at ages 30–69 years (57%), from June to September (48%), and occurring outdoors (64%). These results match the relevant results for the MDS. However, MDS snakebite deaths were equal between males and females. The leg was the dominant site of bite (77%), and the time of reported bite was throughout the day. Of the treated cases, nearly two-thirds (66%) were seen within 1–6 hr, with the remainder seen after six hours. The proportion treated within 1–6 hr improved over time (data not shown). In the fewer studies that attempted to identify the snake species, Russell's viper (*Daboia russelii*) constituted 43%, followed by unknown species (21%), krait (*Bungarus* species) (18%), and cobra (*Naja* species) (12%).

## Snakebite and mortality surveillance

The Government of India relies on reporting via public hospitals to track snakebites and deaths (*Government of India, 2015*). We examined the total bites and deaths available from 2003 to 2015 in Government hospitals and compared these deaths to the MDS in-hospital deaths (*Table 4*). Over this 13-year period, the MDS estimated about 154,000 snakebite deaths in public and private

**Table 3.** Snakebite death rates by state in India for 2001-2014.

| State | Study deaths in MDS | Annual average standardized death rate /100,000 | | | Trend | Estimated deaths for 2001-14 (000) |
|---|---|---|---|---|---|---|
| | | 2001-2004 | 2005-2009 | 2010-2014 | | |
| Higher burden states | 1726 | 5.9 | 6.1 | 6.2 | | 557.4 |
| Andhra Pradesh | 271 | 8.5 | 7.3 | 5.6 | | 82.9 |
| Bihar | 321 | 5.6 | 7.6 | 8.9 | | 101.9 |
| Odisha | 191 | 7.5 | 7.2 | 5.9 | | 40.3 |
| Madhya Pradesh | 195 | 6.7 | 7.7 | 6.0 | | 67.8 |
| Uttar Pradesh | 322 | 5.2 | 5.9 | 6.0 | | 153.6 |
| Rajasthan | 192 | 4.9 | 6.7 | 5.0 | | 52.1 |
| Gujarat | 176 | 4.1 | 4.8 | 5.1 | | 38.8 |
| Jharkhand | 58 | 4.9 | 2.0 | 7.1 | | 20.1 |
| Lower burden states | 1107 | 3.7 | 3.1 | 2.1 | | 249.9 |
| Chhattisgarh | 42 | 6.0 | 6.5 | 2.5 | | 16.8 |
| Jammu & Kashmir | 64 | 5.3 | 7.0 | 0.9 | | 7.0 |
| Tamil Nadu | 176 | 6.1 | 3.4 | 3.0 | | 42.1 |
| Karnataka | 137 | 5.6 | 3.3 | 2.9 | | 33.0 |
| Maharashtra | 147 | 4.2 | 3.7 | 2.6 | | 56.0 |
| West Bengal | 188 | 4.1 | 3.3 | 2.9 | | 42.7 |
| Punjab | 67 | 2.9 | 3.1 | 4.0 | | 14.5 |
| Haryana | 45 | 2.9 | 3.3 | 1.8 | | 9.5 |
| Assam | 27 | 2.8 | 0.7 | 2.1 | | 7.3 |
| Northeastern states | 37 | 2.3 | 0.9 | 0.7 | | 2.4 |
| Kerala | 43 | 1.8 | 1.3 | 0.5 | | 6.5 |
| All other states | 134 | 4.3 | 3.9 | 3.2 | | 12.2 |
| All India | 2833 | 5.1 | 4.9 | 4.5 | | 807.5 |

States are in descending order of annual average death rates for the study period of 2001-2014. We included only the states with populations over 10 million. Andhra Pradesh included Telangana. The Northeastern states include Arunachal Pradesh, Manipur, Meghalaya, Mizoram, Nagaland, Sikkim and Tripura.

hospitals, and the Government reported 15,500 deaths in hospitals, meaning that the routine reporting system captured only 10% of the expected hospital-based deaths. The most complete reporting was in Karnataka which captured 26% of expected hospital snakebite deaths.

## Snakebite prevalence and envenomation

Among the 87,590 snakebites reported in the literature, there were 3329 reported deaths (*Appendix 2—table 1*). We fitted death and bite data from each study to an ordinary least square regression to calculate a case-fatality rate, after removing the extreme outliers. We estimated a crude case-fatality rate of 3.2% for in-hospital cases. Based on mostly cautious assumptions about the ratio of in-hospital to out-of-hospital prevalence of snakebites (*Appendix 1—table 2*), we estimate the total number of snakebites to range from 1.11 to 1.77 million in 2015. Based on 44 hospital studies where 70% of patients sought treatment, were diagnosed with systemic envenomation, and received antivenom, we estimate that the annual number of envenomations is about 0.77 to 1.24 million with the remainder being 'dry bites' or bites by non-venomous species (0.33 to 0.53 million).

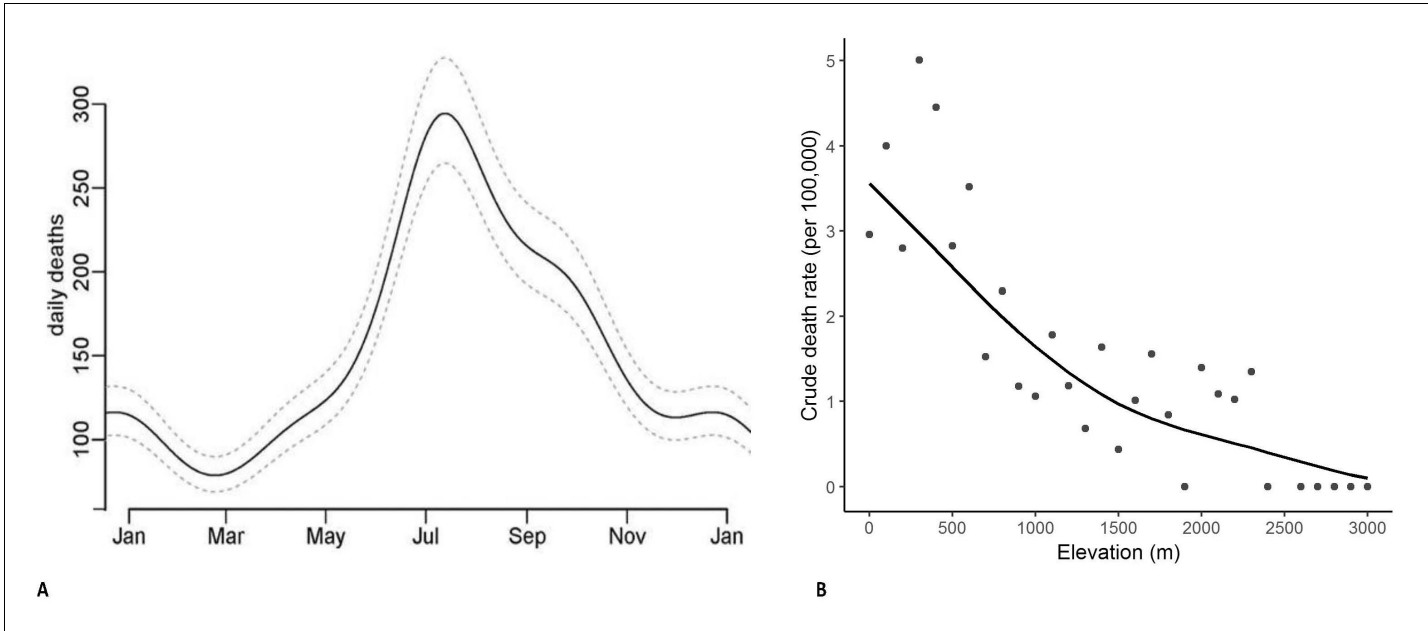

**Figure 2.** Predicted daily snakebite deaths from analysis of seasonality observed in 2001-2014 (Panel A) and snakebite crude death rates by altitude in meters in 2004-2013 (Panel B). Notes: The daily snakebite totals are a composite of all study years from 2001 to 2014. The crude death rates by elevation use the RGI's Sample Registration System population as denominators, and hence are generally lower than the overall rates we apply to the whole of India (using the United Nations death totals, which has the benefit of taking into account undercounts in the SRS data [*Menon et al., 2019*]). However, the relationship of crude death rates with elevation is unaffected by this procedure.

## Discussion

Our nationally representative mortality study documents about 1.2 million snakebite deaths from 2000 to 2019. Most occurred at home in the rural areas. About 70% occurred in eight higher-burden states and half occurred during the rainy season and in low altitude rural areas. While rates of childhood and young adult snakebite mortality have fallen, those in middle age have not. Thus, the average risk of an Indian dying from snakebite before age 70 is approximately 1 in 250, but in some areas, this risk approaches 1 in 100. Over 260 million Indians live in areas of moderate risk of about 1 in 167. More crudely, approximately 1.11–1.77 million bites occur annually with about 70% representing envenomation, and 58,000 dying. While snakebite deaths represent only about 0.5% of the approximately 10 million deaths that occurred in India in 2015, they are nonetheless important, as they are nearly all avoidable.

Many of the features of snakebites and deaths were known or suspected, but few were quantified reliably (*Mohapatra et al., 2011*). Our study's novel contributions are to quantify some of these features, and identify other findings that are relevant to improved epidemiological understanding and to prevention and treatment in snakebite control programs. The map of the snakebite mortality risk (*Figure 1*) highlights 'hot spots' in each state, which are at lower altitude. This reflects not only the more highly populated and the more extensive and intensively farmed arable land at lower altitudes, but also the species and population densities of snake species of medical importance. These snake densities are sometimes very high, particularly in grain agriculture which attracts the largest rodent and amphibian populations that are eaten by snakes (*Whitaker and Captain, 2004*; *Mise et al., 2016*). Focusing on agrarian communities in specific areas which carry the highest risk of mortality, especially during the monsoon seasons, could reduce mortality and morbidity attributable to snakebites. Targeting these areas with education about simple methods, such as 'snake-safe' harvest practices, wearing rubber boots and gloves and using rechargeable torches (or mobile phone flashlights) could reduce the risk of snakebites. Mass distribution of mosquito nets (which also protect against scorpion sting and mosquito-borne diseases) is a relevant strategy that could build upon the National Vector Borne Disease Control Program's efforts to control malaria, kala-azar, and arboviral infections.

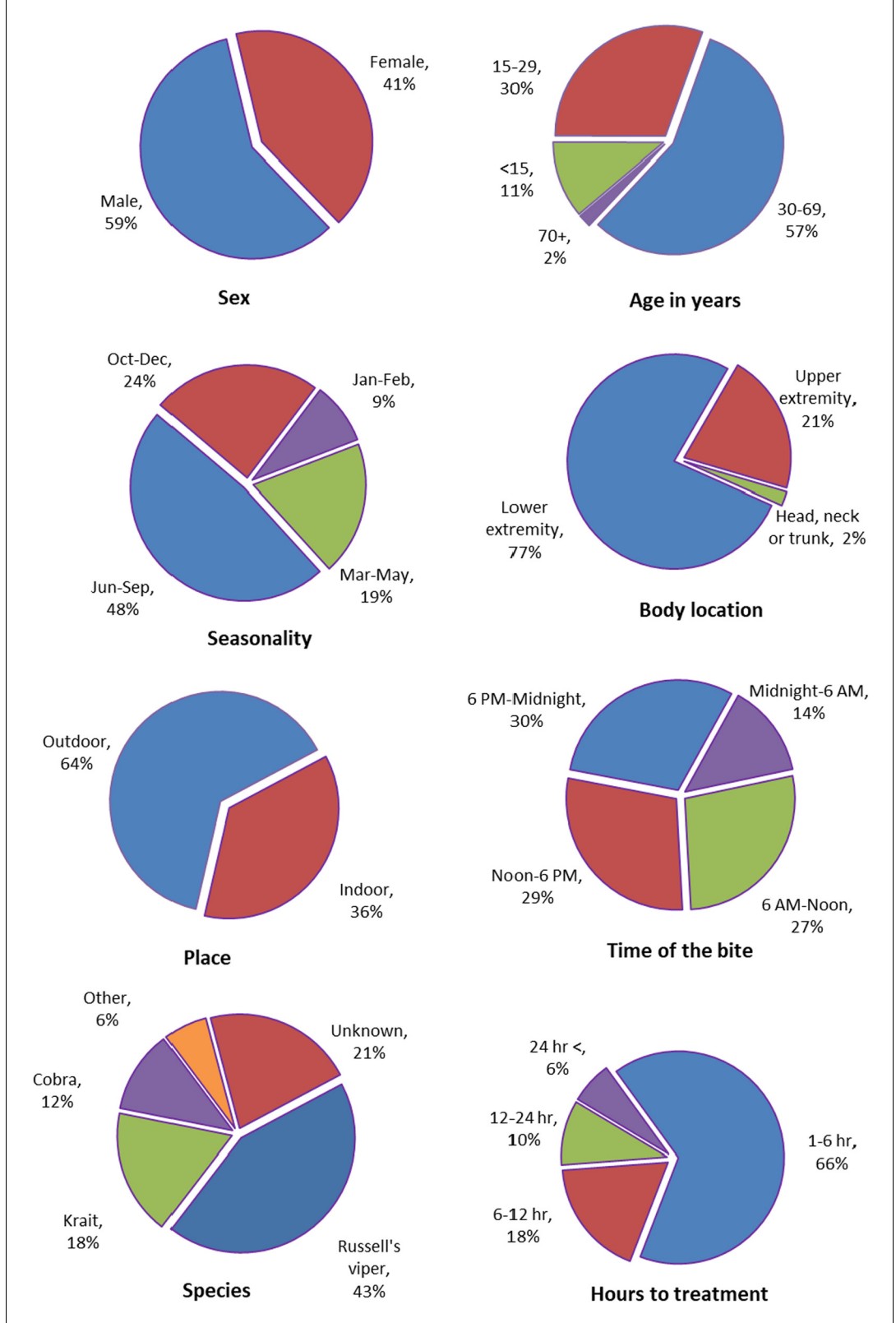

**Figure 3.** Characteristics of snakebites from analysis of 88,000 snakebite events in the published literature.

**Table 4.** Government hospital reports of snakebites and deaths, compared to MDS death totals by state for 2003-2015.

| State | Government reporting[*] | | | MDS estimates | | |
| | No. of bites (000) | No. of deaths (000) | % died in hospital | Total no. of deaths (000) | No. died in hospital (000) | % Government coverage |
| (1) | (2) | (3) | (4) | (5) | (6)=(4)*(5) | (7)=(3)/(6) |
| Higher burden states[†] | 530.0 | 6.9 | 19% | 539.6 | 94.6 | 7% |
| Andhra Pradesh | 251.3 | 1.4 | 16% | 74.1 | 11.6 | 12% |
| Bihar | 20.9 | 0.1 | 16% | 105.4 | 17.0 | 1% |
| Odisha | 76.2 | 1.8 | 29% | 36.9 | 10.9 | 17% |
| Madhya Pradesh | 28.3 | 1.1 | 22% | 64.4 | 14.1 | 8% |
| Uttar Pradesh | 27.8 | 0.6 | 13% | 150.7 | 20.2 | 3% |
| Rajasthan | 71.3 | 1.0 | 16% | 49.6 | 7.8 | 13% |
| Gujarat | 45.7 | 0.8 | 27% | 38.6 | 10.5 | 7% |
| Jharkhand | 8.5 | 0.1 | 12% | 20.0 | 2.5 | 6% |
| Percentage to national | 41% | 45% | | 71% | 61% | |
| Lower burden states | 772.2 | 8.6 | 28% | 219.8 | 59.6 | 14% |
| Chhattisgarh | 16.7 | 0.3 | 17% | 14.3 | 2.4 | 13% |
| Jammu & Kashmir | 18.4 | 0.0 | 26% | 5.9 | 1.5 | 2% |
| Tamil Nadu | 106.6 | 0.5 | 28% | 36.0 | 10.0 | 5% |
| Karnataka | 89.2 | 1.6 | 21% | 28.1 | 6.0 | 26% |
| Maharashtra | 178.7 | 1.2 | 25% | 49.0 | 12.5 | 10% |
| West Bengal | 208.9 | 3.4 | 41% | 38.9 | 15.9 | 22% |
| Punjab | 9.1 | 0.2 | 10% | 14.2 | 1.4 | 13% |
| Haryana | 14.3 | 0.1 | 16% | 8.5 | 1.3 | 11% |
| Assam | 3.6 | 0.1 | 32% | 6.6 | 2.1 | 3% |
| Northeastern states | 11.1 | 0.1 | 25% | 1.8 | 0.4 | 11% |
| Kerala | 37.9 | 0.2 | 27% | 5.2 | 3.5 | 6% |
| All other states | 77.8 | 0.8 | 24% | 11.1 | 2.7 | 32% |
| Percentage to national | 59% | 55% | | 29% | 39% | |
| India | 1302.2 | 15.5 | 22% | 759.4 | 154.2 | 10% |

[*] Government statistics are as published by the Ministry of Health and Family Welfare, Government of India (**Government of India, 2015**).

[†] Higher burden states are those where the snakebite death rate at all ages is above 5/100,000 deaths for the entire study period of 2001-14 as listed in **Table 3**. In cases of number less than 100 deaths, they are listed as 0.0 in thousands.

Our study has implications for better treatment, particularly in the distribution of effective anti-venom to the areas and populations in greatest need. Increased use of antivenom would require tactful cooperation with local traditional healers and ayurvedic practitioners to persuade them to refer severely ill patients for treatment with antivenom, and raising awareness of the effectiveness of antivenom. Government hospitals can make antivenom freely available to snakebite victims (**Whitaker and Whitaker, 2012**). Health services could monitor adverse reactions to antivenom and improve distribution and cold-chain storage, matching supply to places and times of greatest need. Training of local medical staff and emergency responders should be improved so that they can administer antivenom by intravenous injection and also identify and treat early anaphylactic reactions. India has sufficient manufacturing capacity to make large amounts of snake antivenom. Better understanding of the distribution of India's many venomous snake species could help in the development of more appropriate antivenoms. The current Indian polyvalent antivenoms neutralize venom

from only common cobra (*Naja naja*) (there are three other Indian cobra species), common krait (*Bungarus caeruleus*) (seven other krait species), Russell's viper (*Daboia russelii*) and saw-scaled viper (*Echis carinatus*) (*World Health Organization (WHO), 2010*; *Warrell, 2011*; *Warrell et al., 2013*, *Whitaker and Martin, 2015*; *Senji Laxme et al., 2019*). The few studies from healthcare facilities found that antivenom treatment reduced deaths by over 90% (*Appendix 2—table 1*).

We estimate, crudely, that the in-hospital case-fatality rate based on the literature was about 3%. This in part reflects delay in reaching medical care, with only about half of the cases doing so within 6 hr. Public-private partnerships for ambulance services are possible. In some states, an emergency ambulance service equips vehicles with lifesaving equipment and drugs, including antivenom. Ambulances can be summoned in 15 states of India by calling a toll free number (*Gimkala et al., 2016*). In 2014, of 27,509 snakebite patients transported to hospitals within 6 hr, 359 patients died within 48 hr of follow-up. This represents a crude case-fatality rate of 1.3%, below the rate we estimate for in-hospital bites. This ambulance model is relevant to other parts of the country, especially the more remote areas in Bihar, Jharkhand and Odisha, which are not currently covered.

Finally, improved surveillance is required of venomous snake species as well as the human consequences of bites. An enhanced snake species database, hosted in collaboration with agricultural and forest departments, of their habitat details, clear photographs, and geographical distributions is now available as a downloadable Google app (*Indian Snakes, 2019*; *World Health Organization (WHO), 2019d*). There are at least 15 species of snake in India responsible for human deaths, and better information about them would aid control (*Whitaker and Whitaker, 2006*; *Whitaker and Martin, 2015*). We show that public facility-based reporting of deaths captures only 10% of expected deaths in public and private hospitals. While much care in India occurs in private hospitals, disease reporting and surveillance by private facilities is likely to be similarly or even more deficient (*Jha and Laxminarayan, 2009*). The Government of India could designate and enforce snakebite as a 'Notifiable Disease' within the Integrated Disease Surveillance Program. However, since most deaths occur at home, community death tracking through ongoing mortality surveillance will be needed. Both community and facility-based surveillance data are essential to better align interventions to prevent and treat snakebites to their heterogeneously distributed burden. For both hospital and community bites, it would be invaluable to document the circumstances and consequences of snakebites, including morbidity sequelae, with simple questions to investigate the circumstances of each bite or death/disability. This could include data such as use of boots, walking in the dark, sleeping patterns, and other questions. Similar investigative tools for HIV/AIDS have recently been proposed to add to the WHO's standard verbal autopsy (*Bogoch et al., 2018*).

## Limitations of our study

The major source of uncertainty in our estimates of snakebite deaths at national level arises not from random errors, as the MDS has a large sample size and the vital rates used as underlying denominators are reasonably complete (*Menon et al., 2019*), but from the misclassification of causes of death in the verbal autopsy. Earlier evaluations of the MDS showed strong reproducibility of the dual physician-coded verbal autopsies, generally low rates of misclassification in children and young and middle-age adults, and high consistency with relevant hospital or clinical data (*Gomes et al., 2017*; *Aleksandrowicz et al., 2014*; *Menon et al., 2019*). Moreover, two independent physicians agreed about 92% of the time on a diagnosis of snakebite deaths. Snakebite mortality may be under-estimated because the phenomenon of painless, unsuspected nocturnal krait (*Bungarus*) bites resulting in 'early morning paralysis' may not be attributed to snakebite (*Saini et al., 1986*; *Ariaratnam et al., 2008*), but this is likely to be a small bias. Our estimates for some states are uncertain due to small number of deaths recorded annually, which also prevented us from examining yearly spatial clustering patterns.

Our estimates of case-fatality rate and envenomations based on the systematic literature review has obvious limitations. First, the exact species of snake cannot be easily identified, and indeed, there are four species of cobra and eight species of kraits in the country (*Whitaker and Captain, 2004*). In addition, each species varies in the circumstances, seasonal and diurnal variation and types of terrain where bites most often occur. For example, anecdotal experience indicate that most bites from common kraits (Bungarus caeruleus) occur at night while people are sleeping on mats on the floor or ground, in or near home, unprotected by tucked-in mosquito nets (*Ariaratnam et al., 2008*; *Kularatne, 2002*; *Bawaskar and Bawaskar, 2002*). Most bites from saw-scaled vipers (Echis

carinatus) happen either when the snake is stepped upon in bare or sandaled feet at night or when cutting grass by hand with a short sickle. Bites by cobras are divided into circumstances such as defensive bites while stepping on them during planting/harvesting crops, reaching into piles of straw or firewood and predatory bites when the cobra mistakes a human hand or foot for a prey item (*Alirol et al., 2010*). Russell's viper (D. russelii) bites occur during the day, inflicted on farmers in the paddy fields or while hand harvesting peanut plants, or at night when someone walks without using a light and steps on the snake (*Whitaker and Captain, 2004*).

Another major uncertainty from the literature review are in the hospital-based data, given expected problems with publication biases and in the differences between patients seeking or not seeking hospital care. For example, serious cases are more likely to be hospitalized, raising the observed case-fatality rate. However, under various scenarios, our report of 20 to 40 bites per death is greater than the crude estimate of 20 envenomations per death we made earlier (*Mohapatra et al., 2011*). Additional hospital-based surveillance, including tracking the severity of treated cases, could further refine the actual ratio of envenomations to deaths. These uncertainties demand appropriate caution in interpreting our basic estimate of the number of envenomations.

## Conclusion

We conclude that snakebite deaths in India are concentrated largely within limited geographical areas, and involve particular communities during specific seasons. Our identification of the focused geographic and temporal spread of snakebites allows targeted prevention and treatment strategies that could help India to achieve the WHO's goal of halving snakebite death and morbidity rates by 2030. Further use of nation-wide, representative epidemiological studies will be essential to review the success of such control programs.

# Materials and methods

## Data sources

To derive comprehensive and up-to-date estimates of snakebite mortality and prevalence, we collected all possible statistics related to snakebites in India from 2000 to 2015. The main data sources for this study were snakebite mortality data from the Indian Million Death Study (MDS), a systematic review of studies published in the scientific literature, and chronological statistics published by the Ministry of Health and Family Welfare of the Government of India.

## Nationally representative mortality data

The methods, strengths, and limitations of the MDS and key results for various diseases have been extensively reviewed and published (*Aleksandrowicz et al., 2014*; *Gomes et al., 2017*; *Menon et al., 2019*). Briefly, in collaboration with the Registrar General of India, the MDS monitored approximately 23 million people in 3.6 million nationally representative households in India from 1998 to 2014. The Registrar General of India's Sample Registration System (SRS) established three sampling frames for the MDS, which covered years 1993-2003, 2004-2013, and 2014-2023. The SRS randomly selects sampling units based on the 1991, 2001, and 2011 censuses for the respective sampling frames (*Registrar General of India, 2017*). Mortality data used in this study were from 2001-2003, 2004-2013, and 2014, generated from these sampling frames. Every six months, about 900 non-medical surveyors recorded the details of each death that occurred in these households during the preceding six months using a well-validated verbal autopsy instrument (based on the 2012 WHO instrument and including a half-page local language narrative). Each record is converted to an electronic form and randomly assigned to two of 404 trained physicians, who each assign a cause of death using ICD-10 codes. Disagreements in assignment undergo anonymous reconciliation, and persisting differences undergo adjudication by a third physician. We included 2833 snakebite deaths in our study by carefully examining 3020 probable snakebite deaths that either of the two physicians had coded as X20 (venomous snakes), X27 (venomous animals) or X29 (not specific). We followed the same inclusion/exclusion method described in our earlier analyses (*Mohapatra et al., 2011*). Out of the 3020 possible snakebite deaths, there were 2779 (92%) deaths in which both coders initially coded to X20. Review of these yielded no misclassified deaths. Re-examination of the symptoms and physician keywords for 105 deaths that one coder had coded as X20 and other coder as X27 or

X29 revealed that 54 (2%) were snakebite deaths. No misclassified deaths were found in the 136 deaths that one coder had coded X20, X27 or X29 and the other coder had assigned a different ICD-10 code.

## Statistical methods

### Geospatial mapping of deaths

The SRS provided population data for the sampling units for 2004–2013 (*Registrar General of India, 2017*). These population values were partitioned into single-year ages by applying 2011 Census *Registrar General of India, 2011* district-level single-year age structure proportions (by sex and rural/urban setting). District codes from 2011 Census (*Registrar General of India, 2011*) were converted to 2001 codes prior to linking with sampling units given that the 2004–2013 sampling frame used 2001 codes. Sampling units that belonged to districts that split in 2011 used 2001 district codes from the parent districts. All sampling units that belonged to the same district, rural/urban setting and sex shared the same age structure proportions. We further linked the MDS data to these population data at the sampling unit level. Statistical analyses were based on data from 7377 geo-coded sampling units (out of 7597 sampling units), after exclusion of sampling units from the islands.

We derived the spatially-smoothed absolute risks of snakebite mortality in India for 2004–2013. First, using mortality data of ages 0–69 years (by 5-year age group) as the outcome and sampling unit population of the same age range as the offset, we fitted a Bayesian Poisson model to obtain the age- and sex-specific snakebite death rates at the national level. We did not include an intercept in the model, but included age-sex interaction term and time trend (using year 2010 as the reference value) as covariates. This formulation allowed us to obtain the estimated age- and sex-specific national death rates for year 2010. We then used a geostatistical Bayesian Poisson model to estimate the spatially smoothed relative risks of snakebite mortality, by comparing the observed snakebite death rates at each sampling unit versus the national death rate (see Appendix 3 for details). The geostatistical models adjusted for time trends, urban/rural status, female illiteracy in rural areas, altitude, and average of long-term monthly mean temperature. We adjusted for urban/rural status of the sampling unit due to the higher risks of snakebite in rural areas compared to urban areas (*Chaves et al., 2015*). We included female illiteracy in rural areas as a proxy of poverty effects on snakebite mortality, since the poor have higher risk of snakebite (*Harrison et al., 2009*). We used sub-district-level female illiteracy data from the 2011 Indian census (*Registrar General of India, 2011*). We included altitude and long-term monthly temperature as covariates since they affect the occurrence of snakebite (*Chaves et al., 2015*). Altitude data came from the NASA Shuttle Radar Topographic Mission's digital elevation data version 4 (*Jarvis et al., 2008*). Long-term monthly mean temperature came from the University of Delaware's air temperature gridded dataset V5.01, with a 0.5 degree latitude/longitude grid resolution (*Willmott and Matsuura, 2001*). We also included spatial random effects and sampling unit-level random effects in the model. Thus, the spatially smoothed relative risks were the predicted relative risks of snakebite mortality across India. These relative risks were assigned to grid cells that covered the country (see Appendix 3: statistical supplement). Finally, we calculated the absolute risks of snakebite mortality across India by multiplying the spatially-smoothed relative risks (in each grid cell) by the national risks of dying before age 70 years from *Table 1*. National risks of dying used the average of annual risks of dying for 2004–13. The absolute risks represent the risk of dying from snakebite before age 70 years at the grid cell location. Population estimates in high-risk areas were obtained by overlaying the absolute risk surface on the Gridded Population of the World version 4 for year 2015 (*Center for International Earth Science Information Network - CIESIN - Columbia University, 2015*). Further technical explanation on the geostatistical Bayesian model is published (*Brown, 2015*). Appendix 3 provides the model form, equations and implementation.

## Systematic literature review

We performed a systematic review of snakebite studies in India. We searched the literature using a combination of keywords related to the study setting, metrics, treatment, snake species, and geography in Ovid MEDLINE(R), PubMed, Web of Science, and Scopus electronic databases. We selected relevant studies published in the English language from January 1 2000 to September 1 2019, to collect data for understanding case-fatality patterns and important snakebite characteristics in India. In

addition, we hand searched articles that had cited our 2011 publication (*Mohapatra et al., 2011*). Appendix 2 provides the keywords and inclusion and exclusion criteria. The search initially found 1417 snakebite mortality and morbidity studies. After a careful review of titles, abstracts and quality of the studies by three independent reviewers (MB, KP and WS), 78 of the 95 possible studies were included in our analysis. We categorized the 78 studies to four study types: autopsy (seven studies: only deaths), emergency medical services (EMS) (one study of prevalence), hospital (66 studies: both prevalence and deaths) and community (four studies: both prevalence and deaths).

## Mortality rates

We applied the SRS probability of selection sampling weights to the snakebite death frequencies to address urban and rural differences. We calculated snakebite mortality fractions using three-year backward moving averages of weighted snakebite death frequencies for each age, sex, and urban/rural stratum for each state of India. We interpolated the mortality fractions, using standard statistical methods for strata with zero death count (*SAS Institute, 2014*). We applied these mortality fractions to SRS and the India census demographic framework to obtain the snakebite death rates. We then adjusted the death rates (usually upward by slight amounts) to the United Nations Population Division (*United Nations, 2019*) estimated India death totals (*United Nations, 2019*) to obtain the numbers of national and sub-national snakebite deaths. To address the remaining noise from crude death rates, we fitted cubic spline regressions to 2003 to 2014 cause-specific death rates while adjusting snakebite mortality to other causes of death to obtain the final estimates. We obtained estimates for the beginning and end of the period, including years 2001 and 2002 where data used for moving averages were less than three years, by extrapolating the spline curves to cover the overall period of 2000–2015. For comparison of rates across the years, we standardized the death rates to the 2001 census population. We calculated the number of in-hospital and out-of-hospital deaths by multiplying estimated deaths by percentages of study deaths that occurred in-hospital and out-of-hospital, as reported in the MDS. *Appendix 1—figure 1* shows the data sources, data inputs and outcomes.

## Snakebite prevalence estimates

We used an indirect method to estimate snakebite prevalence (which given very short duration of each bite effectively represents incidence) measured in terms of in-hospital and out-of-hospital prevalences. This involved using the estimated MDS hospital deaths divided by case-fatality rate from systematic review of the literature to estimate the in-hospital prevalence and apply a hypothetical relationship between in-hospital and out-of-hospital prevalence to estimate the out-of-hospital prevalence. This is, by necessity, crude but provides some reasonable ranges to estimate the numbers of bites and envenomations in India in recent years. We applied all such prevalence estimates for 2015, as that was the closest year to the last MDS round of 2014. *Appendix 1—table 2* provides details of the calculation and the assumptions.

## Acknowledgements

We thank David Lightfoot for assistance with the literature search and Peter Rodriguez and Leslie Newcombe for data support.

## Additional information

### Competing interests

Prabhat Jha: Reviewing editor, *eLife*. The other authors declare that no competing interests exist.

### Funding

| Funder | Grant reference number | Author |
| --- | --- | --- |
| University of Toronto | | Prabhat Jha |
| Canadian Institutes of Health Research | FDN154277 | Prabhat Jha |

The funders had no role in study design, data collection and interpretation, or the decision to submit the work for publication.

### Author contributions
Wilson Suraweera, Conceptualization, Resources, Data curation, Formal analysis, Validation, Visualization, Methodology, Writing - original draft; David Warrell, Conceptualization, Resources, Supervision, Visualization, Methodology, Writing - review and editing; Romulus Whitaker, Conceptualization, Resources, Validation, Visualization, Writing - review and editing; Geetha Menon, Rashmi Rodrigues, Validation, Visualization, Writing - review and editing; Sze Hang Fu, Formal analysis, Validation, Visualization, Writing - review and editing; Rehana Begum, Data curation, Validation, Project administration, Writing - review and editing; Prabha Sati, Project administration, Writing - review and editing; Kapila Piyasena, Data curation, Formal analysis, Validation, Visualization, Writing - review and editing; Mehak Bhatia, Software, Formal analysis, Writing - review and editing; Patrick Brown, Software, Formal analysis, Validation, Visualization, Methodology, Writing - review and editing; Prabhat Jha, Conceptualization, Resources, Data curation, Supervision, Funding acquisition, Validation, Visualization, Methodology, Writing - original draft, Project administration, Writing - review and editing

### Author ORCIDs
Wilson Suraweera (iD) https://orcid.org/0000-0001-9673-5746
Prabhat Jha (iD) https://orcid.org/0000-0001-7067-8341

### Decision letter and Author response
Decision letter https://doi.org/10.7554/eLife.54076.sa1
Author response https://doi.org/10.7554/eLife.54076.sa2

## Additional files

### Supplementary files
• Transparent reporting form

### Data availability
Data from the Million Death Study (MDS) India cannot be redistributed outside of the Centre for Global Health Research due to legal agreement with the Registrar General of India. However, access to MDS data can be granted via data transfer agreements, upon request to the Office of the Registrar General, RK Puram, New Delhi, India (rgi[dot]rgi[at]nic[dot]in). The public census reports and Sample Registration data can be accessed thorough http://censusindia.gov.in/. Source data for Figure 1 is already explained in Appendix 3. Source data files have been provided for Figure 2 and for Appendix 1-figure 2. The systematic review data used for Figure 3 is already presented in Appendix 2-table 1.

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

## Appendix 1

The role of data sources and snakebite mortality/prevalence estimates for India.

**Appendix 1—table 1.** Snakebite deaths in the present study and mortality estimates by age and gender in 2000-2014.

| Age group/Year | 2001 | 2002 | 2003 | 2004 | 2005 | 2006 | 2007 | 2008 | 2009 | 2010 | 2011 | 2012 | 2013 | 2014 | 2001-14 |
|---|---|---|---|---|---|---|---|---|---|---|---|---|---|---|---|
| All ages | | | | | | | | | | | | | | | |
| Study deaths[*] | | | | | | | | | | | | | | | |
| Total | 199 | 183 | 179 | 190 | 244 | 214 | 225 | 215 | 183 | 200 | 185 | 227 | 214 | 175 | 2833 |
| Male | 120 | 106 | 103 | 110 | 130 | 123 | 115 | 119 | 92 | 115 | 98 | 130 | 113 | 105 | 1579 |
| Female | 79 | 77 | 76 | 80 | 114 | 91 | 110 | 96 | 91 | 85 | 87 | 97 | 101 | 70 | 1254 |
| Rural | 190 | 177 | 172 | 174 | 223 | 207 | 207 | 205 | 172 | 183 | 170 | 210 | 206 | 156 | 2652 |
| Urban | 9 | 6 | 7 | 16 | 21 | 7 | 18 | 10 | 11 | 17 | 15 | 17 | 8 | 19 | 181 |
| In hospital | 40 | 46 | 40 | 32 | 48 | 37 | 49 | 57 | 52 | 38 | 36 | 50 | 52 | 52 | 629 |
| Out of hospital | 159 | 137 | 139 | 158 | 196 | 177 | 176 | 158 | 131 | 162 | 149 | 177 | 162 | 123 | 2204 |
| Estimated deaths (000) [†] | | | | | | | | | | | | | | | |
| Total | 55.0 | 55.3 | 55.8 | 55.6 | 60.8 | 62.7 | 61.0 | 57.4 | 53.8 | 52.4 | 54.9 | 59.2 | 62.3 | 61.2 | 807.5 |
| Male | 29.5 | 29.4 | 29.5 | 28.7 | 31.4 | 32.4 | 31.5 | 29.5 | 27.6 | 26.9 | 28.2 | 30.6 | 32.1 | 31.1 | 418.5 |
| Female | 25.5 | 25.9 | 26.3 | 26.9 | 29.4 | 30.3 | 29.5 | 27.9 | 26.2 | 25.6 | 26.6 | 28.6 | 30.2 | 30.1 | 389.0 |
| Standardized death rate [‡] | | | | | | | | | | | | | | | |
| Male | 5.4 | 5.2 | 5.2 | 4.9 | 5.3 | 5.4 | 5.1 | 4.7 | 4.3 | 4.2 | 4.3 | 4.5 | 4.7 | 4.5 | 4.8 |
| Female | 4.9 | 4.9 | 4.9 | 4.9 | 5.2 | 5.3 | 5.0 | 4.7 | 4.3 | 4.2 | 4.4 | 4.6 | 4.9 | 4.8 | 4.8 |
| Probability of dying (%) [§] | | | | | | | | | | | | | | | |
| Male | 0.40 | 0.40 | 0.39 | 0.37 | 0.41 | 0.42 | 0.40 | 0.36 | 0.33 | 0.31 | 0.32 | 0.34 | 0.36 | 0.35 | 0.37 |
| Female | 0.37 | 0.37 | 0.38 | 0.38 | 0.39 | 0.39 | 0.37 | 0.35 | 0.34 | 0.34 | 0.36 | 0.38 | 0.40 | 0.40 | 0.37 |
| Children age under 15 years | | | | | | | | | | | | | | | |
| Study deaths | | | | | | | | | | | | | | | |
| Male | 45 | 33 | 24 | 38 | 35 | 39 | 32 | 41 | 24 | 27 | 28 | 25 | 22 | 12 | 425 |
| Female | 29 | 20 | 15 | 30 | 40 | 27 | 33 | 25 | 26 | 29 | 25 | 31 | 28 | 14 | 372 |
| Estimated deaths (000) | | | | | | | | | | | | | | | |
| Male | 10.8 | 10.4 | 10.0 | 8.7 | 9.2 | 9.3 | 9.1 | 8.5 | 7.8 | 7.3 | 7.0 | 6.7 | 6.3 | 5.4 | 116.4 |
| Female | 7.9 | 8.3 | 8.7 | 8.5 | 8.6 | 8.4 | 7.8 | 7.3 | 7.0 | 7.3 | 8.2 | 9.3 | 10.1 | 10.1 | 117.8 |
| Age-specific death rate | | | | | | | | | | | | | | | |
| Male | 5.6 | 5.4 | 5.1 | 4.4 | 4.7 | 4.7 | 4.6 | 4.3 | 3.9 | 3.7 | 3.5 | 3.4 | 3.2 | 2.8 | 4.2 |
| Female | 4.5 | 4.7 | 5.0 | 4.8 | 4.9 | 4.7 | 4.4 | 4.1 | 3.9 | 4.1 | 4.6 | 5.3 | 5.8 | 5.8 | 4.8 |
| Ages 15-29 years | | | | | | | | | | | | | | | |
| Study deaths | | | | | | | | | | | | | | | |
| Male | 22 | 30 | 28 | 31 | 32 | 22 | 26 | 20 | 17 | 26 | 23 | 30 | 32 | 18 | 357 |
| Female | 19 | 24 | 19 | 10 | 22 | 19 | 21 | 18 | 21 | 18 | 20 | 21 | 19 | 12 | 263 |
| Estimated deaths (000) | | | | | | | | | | | | | | | |
| Male | 5.6 | 5.7 | 5.7 | 6.0 | 6.2 | 5.9 | 5.3 | 4.6 | 4.0 | 3.9 | 4.5 | 5.4 | 6.3 | 6.5 | 75.5 |
| Female | 4.8 | 4.8 | 4.8 | 5.1 | 4.6 | 4.4 | 4.4 | 4.6 | 4.8 | 4.9 | 4.8 | 4.5 | 4.1 | 3.8 | 64.4 |
| Age-specific death rate | | | | | | | | | | | | | | | |

*Appendix 1—table 1 continued on next page*

*Appendix 1—table 1 continued*

| Age group/Year | 2001 | 2002 | 2003 | 2004 | 2005 | 2006 | 2007 | 2008 | 2009 | 2010 | 2011 | 2012 | 2013 | 2014 | 2001-14 |
|---|---|---|---|---|---|---|---|---|---|---|---|---|---|---|---|
| Male | 3.7 | 3.6 | 3.6 | 3.7 | 3.7 | 3.5 | 3.1 | 2.6 | 2.3 | 2.2 | 2.5 | 3.0 | 3.4 | 3.6 | 3.2 |
| Female | 3.4 | 3.3 | 3.3 | 3.4 | 3.0 | 2.8 | 2.8 | 2.9 | 3.0 | 3.0 | 2.9 | 2.7 | 2.5 | 2.3 | 2.9 |
| **Ages 30-69 years** | | | | | | | | | | | | | | | |
| **Study deaths** | | | | | | | | | | | | | | | |
| Male | 49 | 38 | 45 | 38 | 56 | 55 | 54 | 48 | 44 | 46 | 40 | 65 | 53 | 65 | 696 |
| Female | 28 | 32 | 35 | 35 | 40 | 39 | 51 | 47 | 37 | 34 | 35 | 39 | 49 | 35 | 536 |
| **Estimated deaths (000)** | | | | | | | | | | | | | | | |
| Male | 11.6 | 11.8 | 12.0 | 12.0 | 14.3 | 15.4 | 15.2 | 14.2 | 13.1 | 12.5 | 13.1 | 14.6 | 16.2 | 17.0 | 193.0 |
| Female | 10.7 | 10.9 | 11.1 | 11.4 | 12.8 | 13.5 | 13.2 | 12.4 | 11.6 | 11.3 | 11.8 | 12.8 | 13.8 | 14.0 | 171.2 |
| **Age-specific death rate** | | | | | | | | | | | | | | | |
| Male | 6.0 | 5.9 | 5.9 | 5.7 | 6.6 | 7.0 | 6.7 | 6.1 | 5.5 | 5.1 | 5.3 | 5.7 | 6.1 | 6.3 | 6.0 |
| Female | 5.8 | 5.7 | 5.7 | 5.6 | 6.2 | 6.4 | 6.1 | 5.6 | 5.1 | 4.8 | 4.9 | 5.2 | 5.4 | 5.4 | 5.6 |
| **Age 70 years or above) [¶]** | | | | | | | | | | | | | | | |
| **Study deaths** | | | | | | | | | | | | | | | |
| Male | 4 | 5 | 6 | 3 | 7 | 7 | 3 | 10 | 7 | 16 | 7 | 10 | 6 | 10 | 101 |
| Female | 3 | 1 | 7 | 5 | 12 | 6 | 5 | 6 | 7 | 4 | 7 | 6 | 5 | 9 | 83 |

Notes:

[*] Study deaths were from 2001 to 2014 MDS study rounds.

[†] Estimated deaths were adjusted to United Nations Population Prospects estimated India deaths (***United Nations, 2019***).

[‡] Annual death rates per 100,000 were standardized to 2001 census year population.

[§] The probability of dying from snakebite before age 70 years in the hypothetical absence of other causes.

[¶] Annual deaths above age 70 years were too few to quantify for death rates or totals, but total death estimates for the whole study period in thousands were 33.5 for males and 35.6 for females.

**Appendix 1—table 2.** Expected snakebite prevalence in 2015.

| In-hospital case-fatality rate/ 100 bites | Out-of-hospital to in-hospital hypothetical ratio $K = I(h')/I(h)$ | Hypothetical % who sought hospital treatment 1/ (k+1) | Expected no. of snakebites in 000 | | | No. of envenomations in 000 | No. of dry bites in 000 |
|---|---|---|---|---|---|---|---|
| | | | In-hospital | Out-of-hospital | Total | | |
| (1) | (2) | (3) | (4) | (5) | (6) | (7) | (8) |
| 3.2 | 1.5 | 40% | 442.2 | 663.4 | 1,105.6 | 773.9 | 331.7 |
| 3.2 | 2.0 | 33% | 442.2 | 884.5 | 1,326.7 | 928.7 | 398.0 |
| 3.2 | 3.0 | 25% | 442.2 | 1326.8 | 1,769.0 | 1238.3 | 530.7 |

Notes:

1. We calculated in-hospital case-fatality rates (CFR) (Column 1) from a regression analysis of 66 relevant studies in the systematic literature review (**Appendix 2—table 1**). We excluded the Government's annual health statistics reporting from public hospitals as case-fatality rates calculated from these data were implausibly low and inconsistent. We used an ordinary least square regression to calculate the combined CFR, treating the number of snakebite deaths as the outcome variable and snakebite prevalence as the explanatory variable while excluding outliers. The in-hospital snakebite case-fatality rate (per 100 bites) is:

$$CFR(h) = \frac{D(h)}{I(h)} * 100 \qquad (1)$$

where D(h) represents the number of in-hospital snakebite deaths and I(h) represents the in-hospital snakebite prevalence. The CFR was 3.2% and 95% CI were (2.5, 3.8) (**Appendix 2—table 1**).

2. The MDS study estimates 62,300 snakebite deaths in 2015, of which 22.7% or 14,200 died in hospitals. Inverting formula 1 with the CFR of 3.2 to solve for I(h) yields 442,200 in-hospital snakebite prevalence in 2015 (Column 4).

3. To estimate out-of-hospital prevalence of snakebites (Column 5), we used a hypothetical relationship between in-hospital and out-of-hospital prevalence. If the out-of-hospital to in-hospital prevalence proportion is 'K', then we can express the out-of-hospital snakebite prevalence I(h') as:

$$I\left(h^{'}\right) = K * I(h) \qquad (2)$$

K is an unknown parameter but can also be expressed by 1/(k+1) to represent the proportion of prevalent snakebite cases that would have sought in-hospital treatment. Given the estimated I(h), we determined I(h') by varying the K values. We applied 1.5, 2.0, and 3.0 as plausible K values (Column 2), corresponding to 40%, 33.3% and 25% of cases who sought treatment (Column 3).

4. The sum of I(h) and I(h') or Columns 4 and 5 is the national snakebite prevalence (Column 6).

5. Among 44 studies, an average of 70% of patients received antivenom after a diagnosis of systematic envenomation (**Appendix 2—table 1**). We applied this percentage to obtain the expected number of envenomations in India (Column 7). The remainder were "dry bites" without envenomation (Column 8).

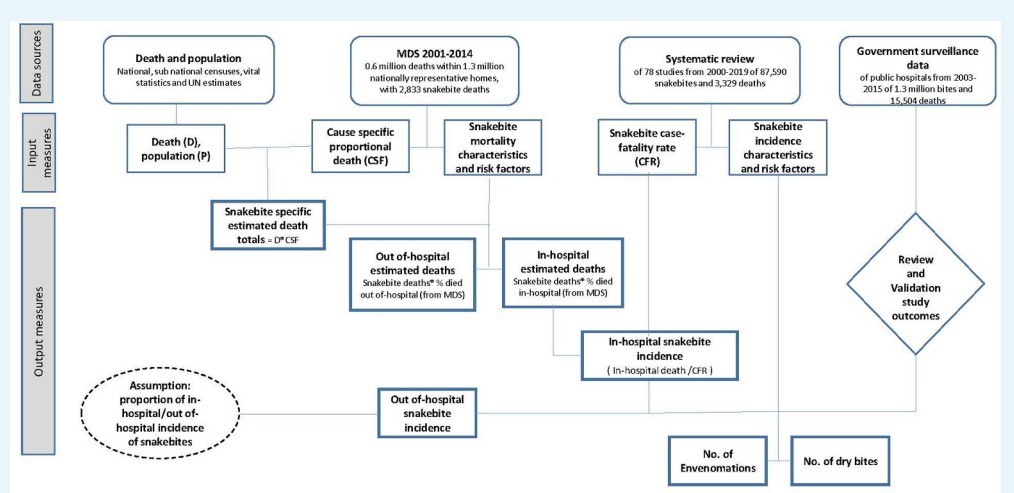

**Appendix 1—figure 1.** The conceptual overview of role of data sources, input measures and study outcomes.

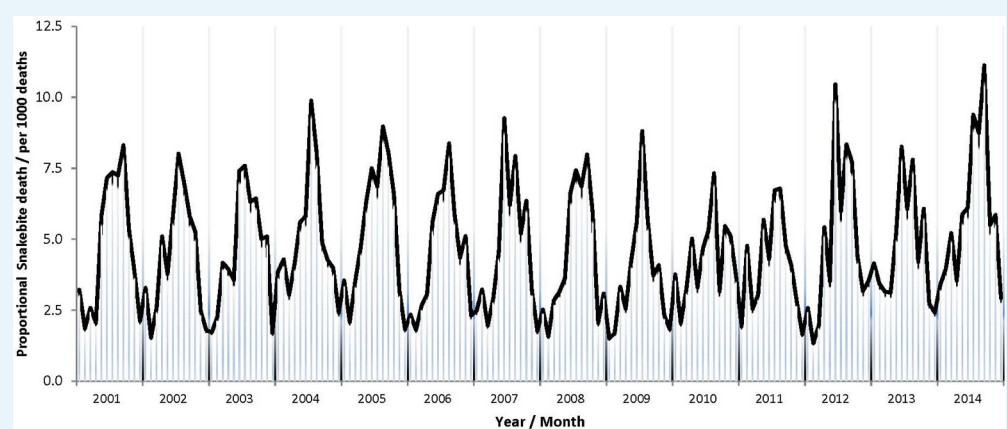

**Appendix 1—figure 2.** Observed seasonality of snakebite deaths in study data in 2001-2014. Note: Proportional snakebite mortality (monthly snakebite deaths to all causes deaths) reported from the 2001 to 2014 rounds of RGI-MDS.

## Appendix 2

Snakebite systematic review search strategy and selection criteria of studies

## Search strategy

We searched Ovid MEDLINE(R), PubMed, Web of Science and Scopus databases for relevant studies from their date of inception from January 1, 2000 to September 1, 2019.

## List of Keyword Categories

**Keyword category**

| Setting | Metrics | Treatment | Snake / Species | Geography / Location |
|---|---|---|---|---|
| Hospital | Death | Antivenom | Venomous | India |
| Survey | Mortality | Treatment | Snakebites | not *Indians, North American |
| Urban or Rural | Incidence | Treated | Bites or Stings | List (of Indian States and Union Territories) |
| Community | Prevalence | Untreated | Envenomation | |
| | Cases | | List (of India venomous snakes from www.indiansnakes.org) | |
| | Case fatality | | | |
| | Mortality rate | | | |
| | Prevalence rate | | | |
| | Incidence rate | | | |

## Inclusion Criteria

We included studies conducted in India that described:

i.   Snakebite cases, both fatal and non-fatal, either exclusively or as a subset of acute poisoning cases,
ii.  Demographic distribution, complications, treatment and outcomes of the snake bite cases.

## Exclusion Criteria

We excluded studies that were:

i.    Secondary reviews,
ii.   Animal or pharmacological studies on therapy drugs or herbal therapies,
iii.  Injury studies on bites and stings by animals other than snake,
iv.   Studies beyond the Indian context,
v.    Biomedical studies that described only the biological mechanism and manifestations of a snakebite,
vi.   Poor quality, failed at the quality assessment.

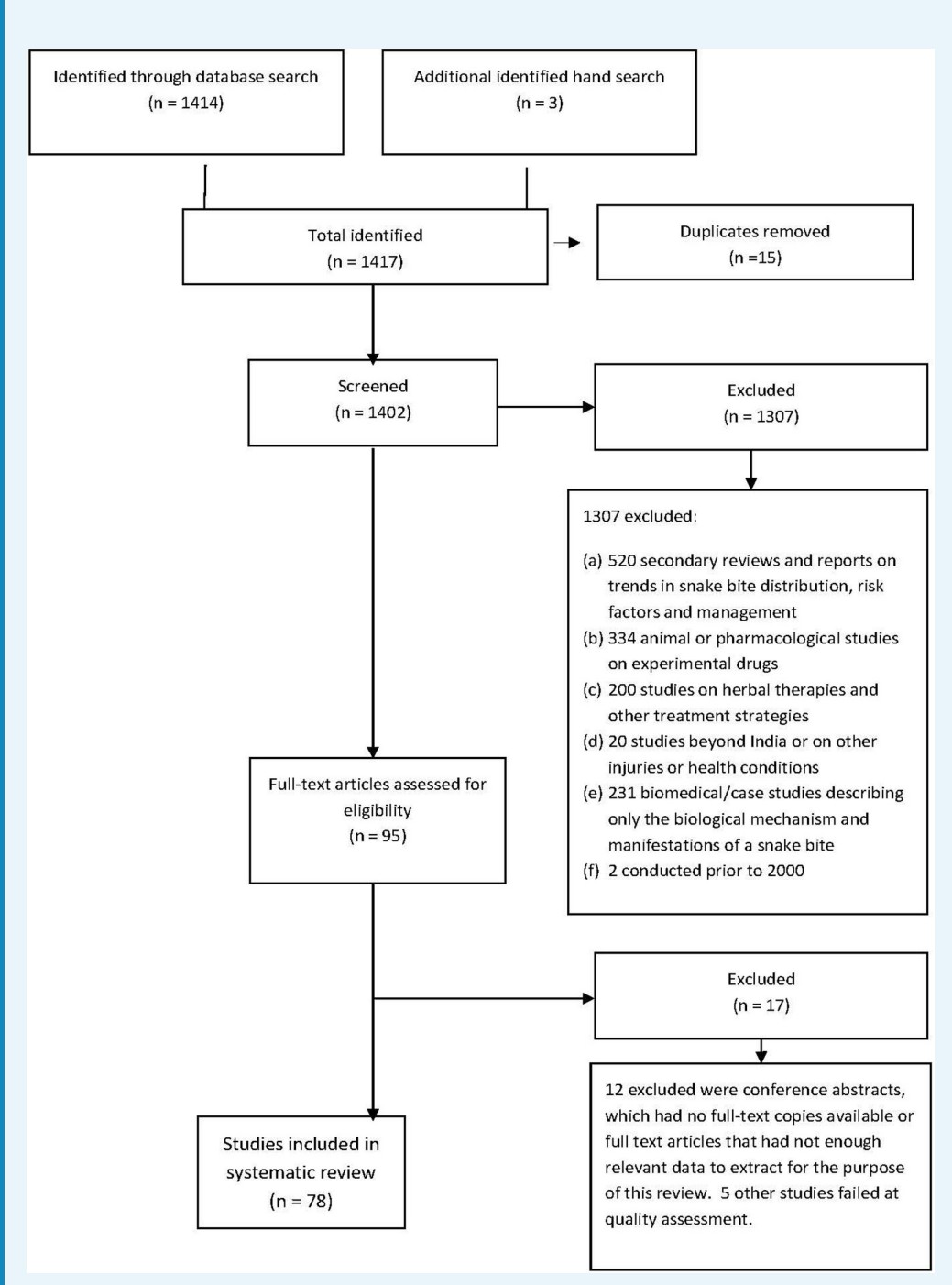

**Appendix 2—figure 1.** Study selection, inclusion and exclusion details.

## List of articles included in systematic review:

### Autopsy studies

1. *Brunda and Sashidhar, 2007*
2. *Chattopadhyay and Sukul, 2011*
3. *Farooqui et al., 2016*
4. *Ghosh et al., 2018*
5. *Kumar et al., 2014*

6. *Tapse et al., 2012*
7. *Tumram et al., 2017*

## Community studies

1. *Armstrong et al., 2019*
2. *Mallikharjuna Rao et al., 2015*
3. *Vaiyapuri et al., 2013*
4. *Venkatesan, 2014*.

## Emergency Medical Services (EMS) studies

1. *Gimkala et al., 2016*

## Hospital studies

1. *Adhisivam and Mahadevan, 2006*
2. *Ahmad and Hussain, 2013*.
3. *Ahmed et al., 2012*
4. *Ahmed et al., 2011*
5. *Ali et al., 2014*
6. *Anil et al., 2010*
7. *Armstrong et al., 2019*
8. *Asawale et al., 2018*
9. *Athappan et al., 2008*
10. *Bakshi, 1999*
11. *Basu et al., 2005*.
12. *Bawaskar et al., 2014*
13. *Bawaskar et al., 2008*
14. *Bawaskar and Bawaskar, 2002*.
15. *Bhalla et al., 2009*
16. *Bhalla et al., 2014*
17. *Chattopadhyay et al., 2004*
18. *Chaudhari et al., 2014*
19. *Chauhan et al., 2005*
20. *Cherian et al., 2013*
21. *Datir et al., 2015*
22. *Deshpande et al., 2013*
23. *Gajbhiye et al., 2019*
24. *Gosavi et al., 2013*
25. *Government of Tamil Nadu, 2008*
26. *Gupt et al., 2015*
27. *Gurudut et al., 2011*
28. *Halesha et al., 2013*
29. *Harshavardhan et al., 2013*
30. *Innah, 2015*
31. *Jayakrishnan et al., 2017*
32. *Kirte et al., 2006*
33. *Korambayil et al., 2015*
34. *Kumar et al., 2018*
35. *Kumar et al., 2013*
36. *Longkumer et al., 2016*.
37. *Mandal et al., 2019*.
38. *Mishra et al., 2019*
39. *Mitra et al., 2015*
40. *Mittal et al., 2012*
41. *Monteiro et al., 2012*
42. *Nagaraju et al., 2015*
43. *Palappallil, 2015*
44. *Panda et al., 2015*
45. *Padhiyar et al., 2018*
46. *Pandey et al., 2016*
47. *Patil et al., 2013*
48. *Patil et al., 2011*

49. *Pore et al., 2015*
50. *Punde, 2005*
51. *Raina et al., 2014*
52. *Ramanath and Naveen Kumar, 2012*
53. *Ramesha et al., 2009.*
54. *Saini et al., 2014*
55. *Sam et al., 2009*
56. *Saravu et al., 2012*
57. *Sarkhel et al., 2017*
58. *Siddique et al., 2015*
59. *Singh et al., 2015*
60. *Singh et al., 2014*
61. *Srivastava et al., 2005*
62. *Swathiacharya et al., 2013.*
63. *Sweni et al., 2012*
64. *Thapar et al., 2015*
65. *Vishwanath, 2019*
66. *Yogesh et al., 2017*

**Appendix 2—table 1. Case-fatality and summary of public health related snakebite characteristics.** (Summary of 78 snakebite studies from 2000 to 2019 included in the systematic review).

| Variable | Autopsy | Community survey | EMS[†] | Hospital | All studies combined | Variable range [**] |
|---|---|---|---|---|---|---|
| **Source of study** | | | | | | |
| **Summary of study outcomes** | | | | | | |
| No. of studies | 7 | 4 | 1 | 66 | 78 | |
| No. of snake bites | 1938 | 1405 | 27509 | 56738 | 87590 | |
| No. of snake bite deaths | 1938 | 131 | 359 | 901 | 3329 | |
| **Case-fatality (per 100 bites)[*]** | | | | | | |
| Crude estimate | n.a. | 9.3 | 1.3 | 1.6 | 3.8 | (1.3, 9.3) |
| Regression estimate | n.a. | 12.2 | 1.2 | 3.2 | | |
| **Summary characteristics of snakebites (n = no. of studies)** | | | | | | |
| **Age (in years) (n = 43)** | | | | | | |
| <15 | 14% | 10% | 11% | 11% | 11.1% | (9.6, 13.8) |
| 15–29 | 24% | 30% | 30% | 32% | 30.4% | (24.0, 32.0) |
| 30–69 | 60% | 58% | 57% | 55% | 56.5% | (54.6, 59.8) |
| 70+ | 2% | 2% | 2% | 2% | 1.9% | (1.7, 2.4) |
| **Sex (n = 66)** | | | | | | |
| Male | 70% | 62% | 56% | 61% | 58.5% | (56.0, 70.2) |
| Female | 30% | 38% | 44% | 39% | 41.5% | (29.8, 44.0) |
| **Season (n = 19)** | | | | | | |
| Summer (March-May) | 18% | n.a. | 19% | 17% | 19.1% | (17.3, 19.5) |
| Monsoon (Jun-Sep) | 51% | n.a. | 47% | 53% | 47.9% | (46.8, 52.8) |
| Post Monsoon (Oct-Dec) | 21% | n.a. | 25% | 21% | 24.1% | (21.1, 24.9) |
| Winter (Jan-Feb) | 10% | n.a. | 9% | 8.8% | 8.9% | (8.8, 9.5) |
| **Place bite happened (n = 16)** | | | | | | |

*Appendix 2—table 1 continued on next page*

*Appendix 2—table 1 continued*

| Variable | Autopsy | Community survey | EMS[†] | Hospital | All studies combined | Variable range [**] |
|---|---|---|---|---|---|---|
| **Source of study** | | | | | | |
| Indoor | 37% | 16% | n.a. | 38% | 36.4% | (15.9, 38.0) |
| Outdoor | 63% | 84% | n.a. | 62% | 63.6% | (62.0, 84.1) |
| **Body location (n = 40)** | | | | | | |
| Lower extremity | 66% | 82% | n.a. | 77% | 76.7% | (66.3, 81.9) |
| Upper extremity | 31% | 16% | n.a. | 21% | 21.2% | (16.2, 30.9) |
| Head, neck or trunk | 3% | 2% | n.a. | 2% | 2.1% | (1.9, 2.8) |
| **Snake species identified (n = 33)** | | | | | | |
| Russell's viper | 71% | n.a. | n.a. | 42% | 43.2% | (42.2, 71.0) |
| Krait | 9% | n.a. | n.a. | 18% | 17.7% | (9.0, 18.0) |
| Cobra | 14% | n.a. | n.a. | 12% | 11.7% | (11.6, 14) |
| Hump nose viper | n.a. | n.a. | n.a. | 4% | 4.0% | (4.2, 4.2) |
| Saw-scaled viper | n.a. | n.a. | n.a. | 2% | 1.7% | (1.8, 1.8) |
| Water snake | n.a. | n.a. | n.a. | 0.4% | 0.3% | (0.4, 0.4) |
| Unknown | 6% | n.a. | n.a. | 22% | 21.3% | (6.1, 21.9) |
| **Time of the bite (n = 18)** | | | | | | |
| 12am-6am | 6% | n.a. | 13% | 23% | 13.6% | (5.6, 22.8) |
| 6am-Noon | 35% | n.a. | 28% | 23% | 27.4% | (23.3, 35.2) |
| Noon-6pm | 39% | n.a. | 29% | 25% | 28.9% | (24.9, 38.9) |
| 6pm-12am | 20% | n.a. | 30% | 29% | 30.1% | (20.4, 30.2) |
| **Hours to treatment (n = 19)[††]** | | | | | | |
| <6 hr | n.a. | n.a. | 100%[‡] | 66% | 65.9% | (23.6, 100) |
| 6–12 hr | n.a. | n.a. | 0% | 18% | 17.9% | (2.7, 36.3) |
| 12–24 hr | n.a. | n.a. | 0% | 10% | 9.8% | (4.1, 33.3) |
| >24 hr | n.a. | n.a. | 0% | 6% | 6.4% | (0.7, 31.9) |
| **Number treated with antivenom (n = 44)[§ ††]** | | | | | | |
| Treated | n.a. | n.a. | n.a. | 70% | 69.7% | (13.3, 100) |
| **Survived by antivenom (n = 19)[¶ ††]** | | | | | | |
| Failed | n.a. | n.a. | n.a. | 6% | 5.6% | (0, 34.6) |
| Survived | n.a. | n.a. | n.a. | 94% | 94.4% | (65.4, 100) |

Notes:

\* Crude case fatality rates are the aggregated number of deaths divided by snakebites. Case-fatality regression estimate was calculated after refining the row data for outliers. After careful assessment of all 66 hospital studies, only 11727 snakebite events followed by 487 deaths from 44 studies were considered for regression estimate (3.17, 95% CI (2.54, 3.79)).

† EMS - '108 call' GVK Emergency Ambulance Services in India (***Gimkala et al., 2016***). EMS data is for 2014 covering 12 states in India (Andhra Pradesh, Chhattisgarh, Dadra and Nagar Haveli, Daman and Diu, Goa, Gujarat, Himachal Pradesh, Karnataka, Meghalaya, Tamil Nadu, Telangana, Uttarakhand). Out of 359 EMS deaths, 168 died before reaching hospital and the remainder died after 48 hr follow up in hospitals.

‡ Time from EMS service call to transport to hospital.

§ Patients who received antivenom at hospital after diagnosis of systematic envenoming.

¶ Survived by antivenom was calculated by dividing the deaths or survivors by number treated with antivenom. n.a. - Not available or not relevant.

** Variable range (minimum, maximum) observed in study groups. When available for only one study group, range within the observed studies in that group.

†† Pooled estimates for 'Hours to treatment', 'Number treated with antivenom' and 'Survived by antivenom' were from hospital studies only.

**Appendix 2—table 2.** Systematic review summary of state coverage by years of 78 studies.

| | State | No. of studies | Total no. of snakebites | Total no. of snakebite deaths | Sources of studies | | | | Publication year/number of studies | | | | | | | | | | | | | | | | | | |
|---|---|---|---|---|---|---|---|---|---|---|---|---|---|---|---|---|---|---|---|---|---|---|---|---|---|---|
| | | | | | Autopsy | Community survey | EMS | Hospital | 1999 | 2002 | 2004 | 2005 | 2006 | 2007 | 2008 | 2009 | 2010 | 2011 | 2012 | 2013 | 2014 | 2015 | 2016 | 2017 | 2018 | 2019 |
| 1 | Andhra Pradesh | 4 | 6283 | 1457 | 1 | 1 | 1 | 1 | | | | | | 1 | | | | | | | 1 | 1 | 1 | | | |
| 2 | Bihar | 3 | 1171 | 29 | | 1 | | 2 | | | | | | | | | | | | | | | 1 | | | 2 |
| 3 | Chandigarh | 4 | 297 | 6 | | | | 4 | | | | | | | | 1 | 1 | | 1 | | | | 1 | | | |
| 4 | Chhattisgarh | 1 | 2084 | 52 | | | 1 | | | | | | | | | | | | | | | | | | | 1 |
| 5 | Dadra and Nagar Haveli | 1 | 384 | 1 | | | 1 | | | | | | | | | | | | | | | | 1 | | | |
| 6 | Daman and Diu | 1 | 12 | | | | 1 | | | | | | | | | | | | | | | | 1 | | | |
| 7 | Goa | 1 | 244 | | | | 1 | | | | | | | | | | | | | | | | 1 | | | |
| 8 | Gujarat | 1 | 3628 | 91 | | | 1 | | 1 | | | | | | | | | | | | | | | | | |
| 9 | Haryana | 1 | 17 | | | | | 1 | | | | | | | | | | | | | 1 | | | | | |
| 10 | Himachal Pradesh | 4 | 1442 | 28 | | | 1 | 3 | | | | | | | | 1 | | | | | 2 | | 1 | | | |
| 11 | Jammu and Kashmir | 1 | 10 | | | | | 1 | | | | | | | | | | 1 | | | | | | | | |
| 12 | Jharkhand | 1 | 356 | 19 | | | | 1 | | | | 1 | | | | | | | | | | | | | | |
| 13 | Karnataka | 20 | 5281 | 139 | 1 | | 1 | 18 | | | 1 | | | | 2 | | | 1 | 4 | 4 | 1 | 3 | 1 | 1 | 1 | 1 |
| 14 | Kerala | 5 | 3169 | 131 | | | | 5 | | | | | | | | | | | | | | 3 | | 1 | 1 | |
| 15 | Maharashtra | 18 | 4884 | 432 | 2 | | | 16 | 1 | 1 | | 1 | 1 | | 1 | | | 1 | 4 | 2 | 3 | 2 | 1 | | | |
| 16 | Meghalaya | 1 | 13 | | | | 1 | | | | | | 1 | | | | | | | | | | | | | |
| 17 | New Delhi | 1 | 62 | | | | | 1 | | | | 1 | | | | | | | | | | | | | | |
| 18 | Odisha | 2 | 101 | 33 | | | | 2 | | | | | | | | | | | | 1 | | 1 | | | | |
| 19 | Pondicherry | 1 | 50 | 9 | | | | 1 | | | | | 1 | | | | | | | | | | | | | |
| 20 | Tamil Nadu | 7 | 51,198 | 371 | 2 | 2 | 1 | 4 | | | | | | | 3 | | | | | 1 | 1 | | 1 | | | 1 |
| 21 | Telangana | 1 | 3956 | 92 | | | 1 | | | | | | | | | | | | | | | | 1 | | | |
| 22 | Uttar Pradesh | 4 | 249 | 90 | 1 | | | 3 | | | | | | | | | | | 1 | 1 | | 1 | 1 | | | |
| 23 | Uttarakhand | 1 | 329 | 4 | | | 1 | | | | | | | | | | | | | 1 | | | | | | |
| 24 | West Bengal | 6 | 2370 | 345 | 2 | | | 4 | | | | 1 | | | | | | 1 | | 1 | | | | 1 | 1 | 1 |

Appendix 2—table 2 continued

| State | No. of studies | Total no. of snakebites | Total no. of snakebite deaths | Sources of studies | | | | | Publication year/number of studies | | | | | | | | | | | | | | | |
| | | | | Autopsy | Community survey | EMS | Hospital | 1999 | 2002 | 2004 | 2005 | 2006 | 2007 | 2008 | 2009 | 2010 | 2011 | 2012 | 2013 | 2014 | 2015 | 2016 | 2017 | 2018 | 2019 |
| Total | 90 | 87,590 | 3329 | 7 | 4 | 12 | 67 | 1 | 1 | 1 | 4 | 2 | 1 | 4 | 3 | 1 | 4 | 8 | 9 | 9 | 13 | 15 | 4 | 4 | 6 |

Note:
The 78 studies we reviewed were published over 20 year's period and data are corresponding to 1999–2019 from 24 different states. Except in a few studies, data of individual studies were confined to a single state. Therefore, possibility of overlapping of study data within states would be minimal and we verified the areas when several studies were conducted in the same year within the same state. One study from 1999 was included because no other studies were found around the year of 2000.

**Appendix 3**

## Statistical supplement to snakebite analysis

### Introduction

Here, we provide further details on distribution of MDS sampling units, the Bayesian statistical methods, and model validations of the estimated absolute risk of the study.

### The sampling units

The India's Sample Registration System (SRS) is based on a system of dual recording of births and deaths in fairly representative sample units spread all over the country. In the 2004–13 sampling frame, there were 7597 sampling units, and 7416 of them were geocoded. Excluding the islands, there were 7377 geocoded sampling units (*Appendix 3—figure 1*).

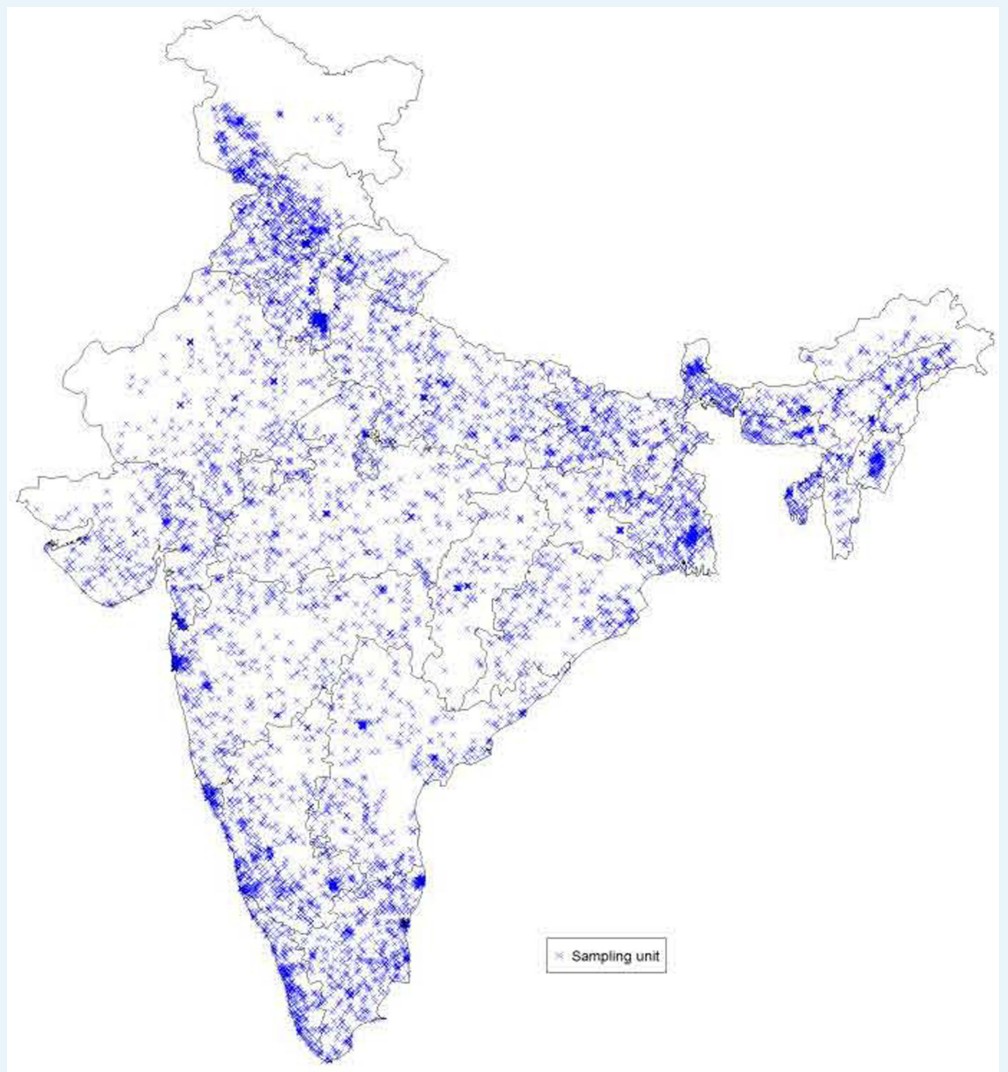

**Appendix 3—figure 1.** Locations of geocoded sampling units from the Sample Registration System (SRS) in 2004–13, excluding sampling units for islands.

## Statistical models

The statistical models were fitted using Bayesian inference, and posterior distributions were computed using the INLA methodology from inla *Rue et al., 2009* as explained in the following sections.

## Spatially smoothed relative risks of snakebite mortality

A Poisson regression model with a spatial random effect was used for the spatial analyses, a model which is a form of Generalized Linear Geostatistical Model described by *Diggle and Ribeiro, 2006*. The approach is broadly similar to the methods used by *Bhatt et al., 2015* for estimating malaria prevalence in Africa, and by *Jiang et al., 2014* for assessing the contribution to cancer incidence of ambient radiation near a nuclear generating facility in Canada.

Writing $Y_{it}$ as the outcome in sampling unit $i$ (located at the spatial coordinates $s_i$) at time $t$, the model is as follows.

$$Y_{it} \sim Poisson(\lambda_{it} E_{it})$$
$$log(\lambda_{it}) = X(S_i, t)\alpha + f_t + U(S_i) + Z_i$$
$$Z_i \sim N(0, \tau^2)$$
$$U(S_i) \sim N(0, \sigma^2)$$
$$cor[U(s+h), U(s)] = \rho(\|h\|\phi)$$

$E_{it}$ is the expected number of snakebite death (described below). The (log-transformed) relative risk $\lambda_{it}$ depends on the spatially referenced covariates $X(s_i, t)$, a non-linear time trend $f$, the spatial random effect $U(s_i)$, and sampling unit-level effect $Z_i$.

- The spatial explanatory variable $X(s_i, t)$ includes an intercept, a rural-urban indicator variable, the sub-district-level proportion of female illiteracy in rural areas from the 2011 census, altitude, and long-term monthly mean temperature. We applied change points for altitude at 400 meters and long-term monthly mean temperature at 20°C based upon exploratory analysis of the relationship between these covariates and the relative risks of snakebite mortality.
- The time trend $f$ is a non-parametric effect (a wiggly line) modelled as a second order random walk, with year 2010 as the reference value.
- The spatial random effect $U(s_i)$ is a Gaussian random field with a Matern spatial correlation function $\rho$ specifying how correlation decreases with distance $h$, depending on the value of a range parameter $\phi$. The range parameter determines how rough or smooth the relative risk surface is.
- The unit-level effect $Z_i$ is spatially independent, and unlike the spatial effect $U(s_i)$, two sampling units in close proximity will have unrelated values of $Z_i$. The $Z_i$ can be thought of as accounting for short-scale spatial variation or sampling-unit-level risk factors not included in the model as covariates.

The absolute risks mapped in main text *Figure 1* is the posterior median of the spatial relative risk $\exp[X(s, t)\alpha + U(s)]$ in 2010 (since $f_{2010} = 0$) multiplied by the national value for risk of dying before age 70 years.

The expected counts $E_{it}$ were obtained by computing age-sex-specific rates (by 5 year age group) using a non-spatial Poisson regression model with the age-sex specific death counts as the response, the population of the age-sex group in the study as an offset, and a linear time trend (with reference at year 2010). The estimated rates were multiplied by each sampling unit's age-sex population in each year and summed over groups to produce each unit's yearly expected count.

## Implementation

As there are 7377 geocoded sampling units, and 6587 different spatial locations $s_i$, model fitting is computationally intensive and an approximation to the spatial covariance matrix is necessary. The Markov random field approximation from *Lindgren et al., 2011* is used here, and implemented in the geostatsp package (*Brown, 2015*) for the R statistical programming

language (*R Development Core Team, 2018*), which in turns calls the inla software (*Rue et al., 2009*). A full description of the methodology is published *Brown, 2016*. Although there are other methods available for fitting models of this type, the task is complex and computationally demanding and there is currently no rival to the Bayesian methodology in the inla software for fitting spatial models with non-Normal responses.

Bayesian inference requires specifying prior distributions, and spatial models are particularly susceptible to producing spurious results from ill-chosen priors for the spatial parameters $\phi$ and $\sigma$. Here the penalized complexity prior distributions from *Simpson et al., 2017* are used, priors which discourage a spatial effect (wanting $U(s)$ flat and close to zero) unless the data indicate a clear preference for a spatial model. Following *Simpson et al., 2017*, the standard deviations $\sigma$ and $\tau$ have exponential priors, as does the scale parameter $1/\phi$. The prior median for $\phi$, the distance beyond which the correlation between two locations is under 10%, is $500km$ or roughly one sixth of the distance across India. The prior medians of $\sigma$ and $\tau$ are both $\log(2)$, a value at which a one standard deviation increase in $U(s_i)$ or $Z_i$ doubles mortality risk.

## Model validation

### Snakebite crude death rates

A map of the crude death rates of snakebite at the sampling units is provided in *Appendix 3—figure 2*. Overall, areas with higher crude death rates correspond to areas with higher absolute snakebite risks (in Main text: *Figure 1*). It should be noted that the crude death rates have not accounted for the other factors that were included as covariates in the spatial Poisson regression model.

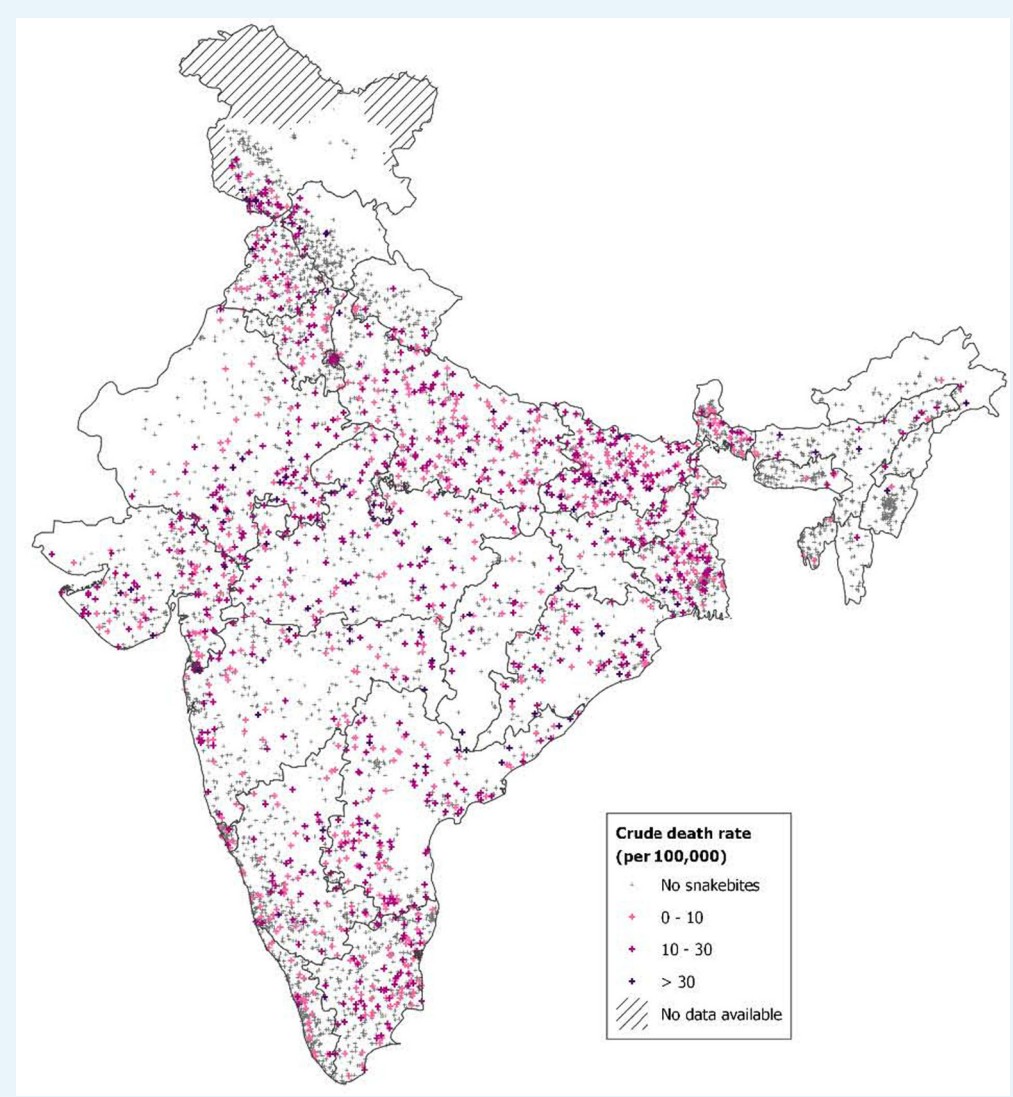

**Appendix 3—figure 2.** Snakebite crude rates in 2004–13.

## Uncertainty of absolute snakebite risks

Uncertainty of the absolute snakebite risks can be depicted by the 95% credible interval of the posterior distribution for each grid cell (*Appendix 3—figure 3a-b*). For comparison, the estimated median absolute snakebite risks is also provided (*Appendix 3—figure 3c*).

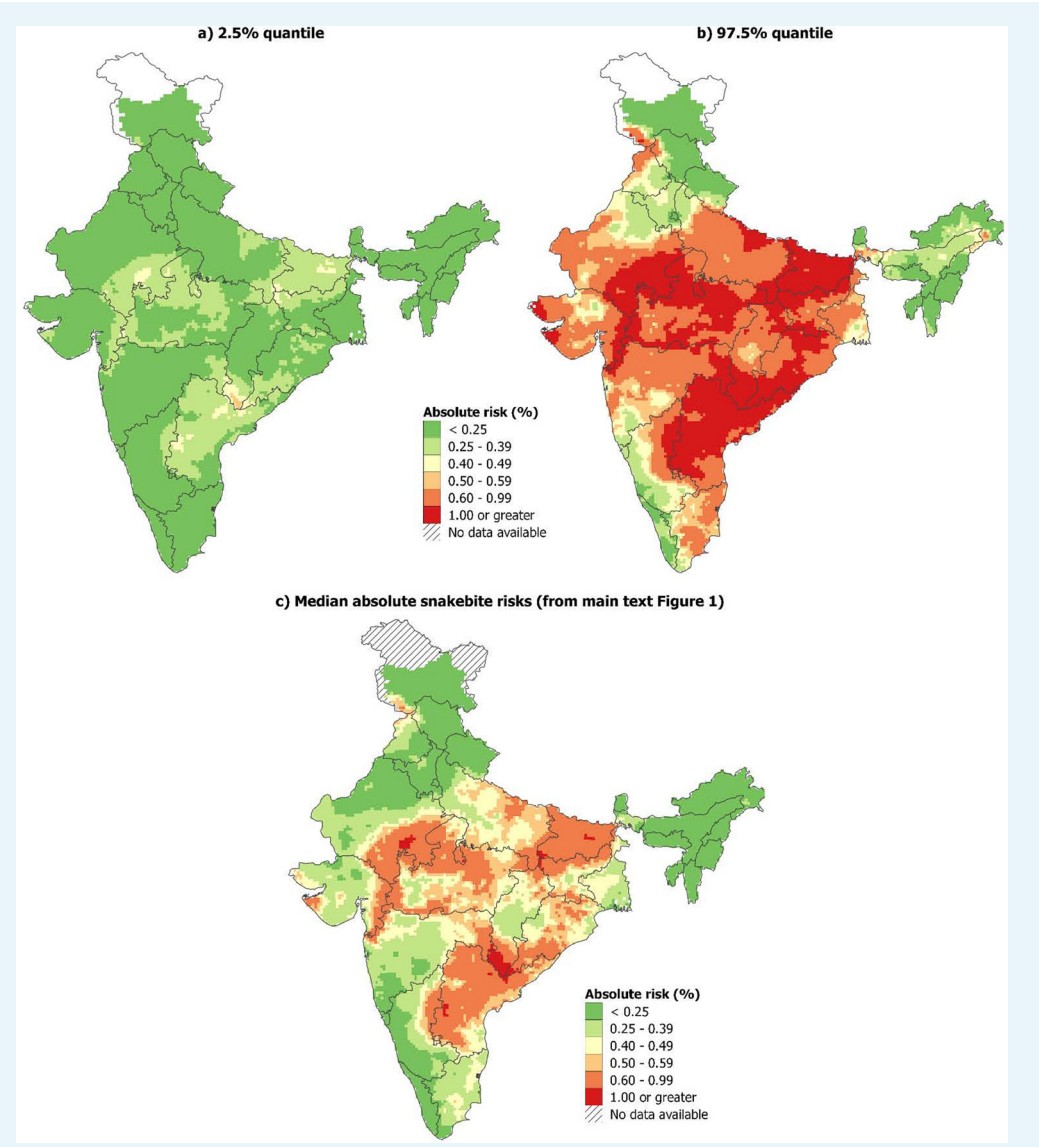

**Appendix 3—figure 3.** 95% credible intervals of the absolute risks and median absolute risks of snakebite deaths in India, 2004-13. (**a-b**) 95% credible interval of the absolute risk estimates, (**c**) median absolute snakebite risks (from main text **Figure 1**).

## Covariate parameter estimates

Here, we provide the parameter estimates for covariates included in the Bayesian spatial Poisson model (*Appendix 3—table 1*). Urban/rural status, female illiteracy in rural areas, altitude above the change point of 400 m, and temperature below the change point of 20°C showed statistically significant association with snakebite mortality risk. Female illiteracy in rural areas and increased temperature below 20°C were associated with higher risks of snakebite mortality, and vice versa for urban status and increased altitude at attitude level above 400 m. For time trend, snakebite mortality risks showed a decreasing trend over time (in terms of year, *Appendix 3—figure 3*).

**Appendix 3—table 1.** Parameter estimates for covariates in the geostatistical Bayesian Poisson model.

| | Relative risk of snakebite mortality | | |
|---|---|---|---|
| | **0.5 quantile** | **0.025 quantile** | **0.975 quantile** |
| (Intercept) | 1.011 | 0.456 | 1.705 |
| Urban vs rural | 0.280 | 0.229 | 0.338 |
| Female illiteracy in rural areas | 1.427 | 1.276 | 1.596 |
| Altitude (below 400 m) | 1.905 | 0.812 | 4.392 |
| Altitude (above 400 m) | 0.508 | 0.305 | 0.824 |
| Temperature (below 20°C) | 4.774 | 1.658 | 15.764 |
| Temperature (above 20°C) | 1.170 | 0.502 | 2.641 |
| range/1000 ($\phi$/1000) | 442.001 | 244.373 | 847.835 |
| sd of spatial random effect ($\sigma$) | 0.587 | 0.442 | 0.796 |
| sd of random walk two for year | 0.003 | 0.001 | 0.007 |
| sd of sampling unit effect ($\tau$) | 0.556 | 0.466 | 0.655 |

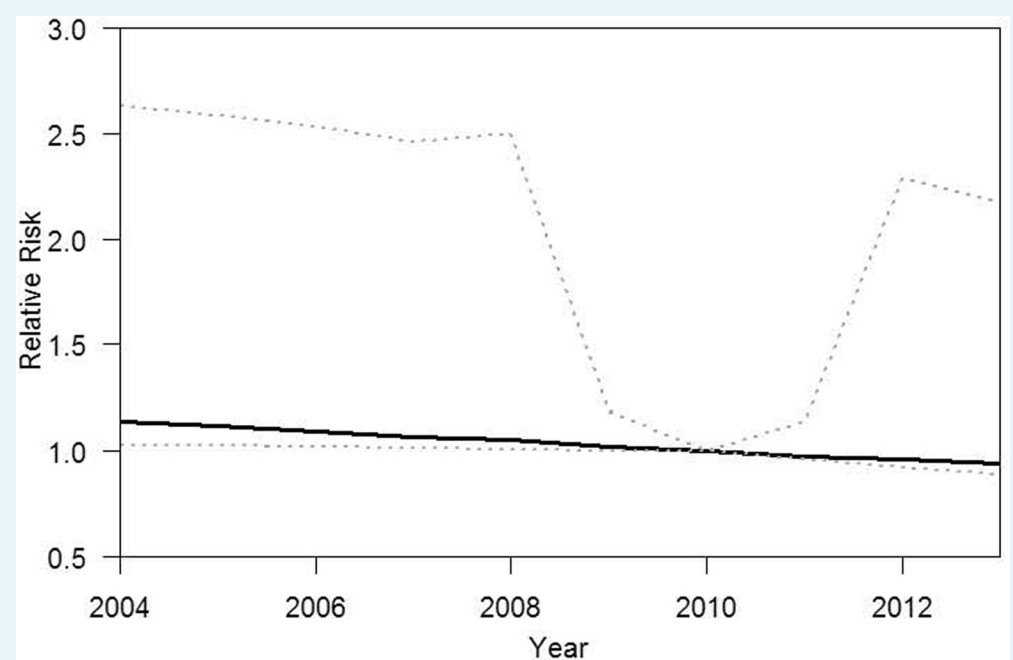

**Appendix 3—figure 4.** Non-linear effects of year estimated using second order random walk.

