## [Decision Letter]

Thank you for submitting your article "Trends in snakebite mortality in India from 2000 to 2019 in a nationally representative mortality study" for consideration by *eLife*. Your article has been reviewed by two peer reviewers, and the evaluation has been overseen by a Reviewing Editor and Neil Ferguson as the Senior Editor. The reviewers have opted to remain anonymous.

The reviewers have discussed the reviews with one another. Unfortunately, they both feel that there is insufficient detail about the methods for them to make a decision about the paper. If you are interested in preparing a revised submission for their consideration, then I have made a list of their concerns which could help you.

1) Please clarify which data sources *and* models were used to estimate the different quantities, including more detail in the Materials and methods of how exactly the different data sources were used/combined. A flow-diagram showing the role of the different sources and models would be sufficient.

2) Many of the statistical model outputs are reported without referring to them as "estimates" – please update wording to make clear these are estimated quantities.

3) Regarding "…has the benefit of correcting for the slight undercounts of deaths…". Please clarify which aspect of Menon et al., 2019, suggests this, and clarify how it is known that increasing these figures is appropriate, and not an overestimate?

4) Your estimates of the risk in each year are from data in the MDS from each of those years – with the population for those years. Though, you standardize each value to the 2001 census (subsection “Trends in snakebite mortality and its geographic and temporal patterns”, second paragraph; Table 1). Should these values not be standardized to the population for each year, so that the combined relative risk and populations give a reasonable estimate of the total deaths in that year?

5) You state that "We found definite seasonality… using a Poisson model…". Did you have a time-series component to the Poisson model that identified seasonality? Or, should these two points be separate? This sentence is unclear.

6) Please provide further details of the "regression analysis that excluded outliers". A) What kind of regression, and B) how were outliers identified, and on what basis were they excluded?

7) Please also provide a figure corresponding to Figure 1 showing the spatial uncertainty in the estimated risks (perhaps in the Appendix).

8) Could you please clarify in the Introduction the purpose of the systematic literature review. It currently states that "We further quantify the levels of envenomations based on…". It appears as though this review adds details on the specific causes, bite locations, and treatment/response to envenomations, while also supporting estimates from the initial analyses.

9) Subsection “Trends in snakebite mortality and its geographic and temporal patterns”, second paragraph: Are the estimates of 55,000 and 61,000 over the same period of time? They are stated as "2001-2" and "2014".

10) Please justify the inclusion of "female illiteracy" in the Poisson model.

11) There is a lack of general consistency in precision of quantitative estimates reported and a general difficulty in differentiating reported numbers versus estimates. Some numbers are reported in general terms "Approximately" etc., whereas others are reported with greater precision, with corresponding uncertainty. While the use of approximately is sufficient in narrative contexts, there are imprecise numbers that do not have a precise version elsewhere in the manuscript. The mixture of reporting of numbers means it is unclear which numbers are estimates, which are reporting of summary statistics from the million deaths studies, and which are reporting of alternative figures. Clearly differentiating estimates from aggregate reports form the study is critical. Table 2 in particular is confusing, and is lacking any uncertainty reporting. You are recommended to use the GATHER statement https://journals.plos.org/plosmedicine/article?id=10.1371/journal.pmed.1002056 as a way of critiquing the current methods and results summarization, as this provides a good checklist for many of the areas that are lacking in this article.

12) What is the nature of the representativeness of the Million Deaths Study? This is particularly important for the geospatial analysis since it is not particularly apparent what the spatial distribution of surveyed locations is. A map of locations, or administrative units should be supplied should permissions allow. Even when not, such as some DHS surveys, protocols are reported and summaries presented so that spatial variation can be appreciated. While references are provided in the article, pertinent information should be supplied in the main manuscript, or appendix to flesh this out.

13) No references or discussion of the provenance of covariates used in the geospatial model are provided. Where do these numbers come from? What is their spatial resolution? How is uncertainty propagated, where relevant? For the local estimates, what was the source of the corresponding local population estimates? If census statistics, how were these translated to the gridded resolution of Figure 1?

14) What are the implications of potential bias in estimates of using mainly hospital derived cfrs and envenomings where there is likely a more urban bias to rural populations, where the nature of healthcare access and treatment seeking behavior could be different. This could be accentuated as different snake species will differentially impact populations due to differences in life history traits altering how they interact with humans. There is not much discussion as to how (a) these forms of bias may exist in the data and (b) what is the expectation of bias in quantitative estimates. The amenable nature of snakebite mortality means that where the potential for fatality to be averted (in a clinical context) may not be equally available for all participants in the MDS, or rural populations more generally, the literature derived CFR might be of limited external validity.

15) The reporting of the geostatistical model is insufficient. It is unclear what the form of the model is and what the equations defining the relationships are. No validation of the model statistics are provided either quantitatively, or in visual reference to a map of snakebite locations. No uncertainty is provided for the estimates. Similar critiques are associated with the use of spline estimates – how was uncertainty propagated here?

---

## [Author Response]

The reviewers have discussed the reviews with one another. Unfortunately, they both feel that there is insufficient detail about the methods for them to make a decision about the paper. If you are interested in preparing a revised submission for their consideration, then I have made a list of their concerns which could help you.1) Please clarify which data sources AND models were used to estimate the different quantities, including more detail in the Materials and methods of how exactly the different data sources were used/combined. A flow-diagram showing the role of the different sources and models would be sufficient.

We have added a detailed flow-diagram to illustrate our methods to Appendix 1.

2) Many of the statistical model outputs are reported without referring to them as "estimates" – please update wording to make clear these are estimated quantities.

We have updated the wording to make clear which are estimates (which are generally robust application of proportions to national total or overall snakebite deaths, given the core mortality data are randomly selected from the whole of India).

3) Regarding "…has the benefit of correcting for the slight undercounts of deaths…". Please clarify which aspect of Menon et al., 2019, suggests this, and clarify how it is known that increasing these figures is appropriate, and not an overestimate?

Now inserted: **“**We applied the age- and sex-specific proportion of snakebite deaths to total deaths as estimated by the United Nations Population Division (UN) for India [United Nations, 2019] to estimate national death rates by age and sex, as well as absolute totals for each year (Table 1). […] The use of the UN death totals adjusts for these possible undercounts, and provides a plausible national total for each year.”

In addition, various demographic groups, including the United Nations Population Division have conducted reviews about the completeness of death reporting. Our procedure of using UN death totals and MDS proportions has been extensively documented and peer-reviewed in past MDS papers. We have added the above text with several more references to buttress the above point. Over estimates are unlikely as long as the proportion of missing deaths was randomly distributed.

4) Your estimates of the risk in each year are from data in the MDS from each of those years – with the population for those years. Though, you standardize each value to the 2001 census (subsection “Trends in snakebite mortality and its geographic and temporal patterns”, second paragraph; Table 1). Should these values not be standardized to the population for each year, so that the combined relative risk and populations give a reasonable estimate of the total deaths in that year?

Thank you. We have clarified the age-specific rates used to calculate annual totals from the overall age-standardized rates (to take into account changes in age) over the 2001-14 period.

5) You state that "We found definite seasonality… using a Poisson model…". Did you have a time-series component to the Poisson model that identified seasonality? Or, should these two points be separate? This sentence is unclear.

Thank you. We have re-written to state exactly this.

6) Please provide further details of the "regression analysis that excluded outliers". A) What kind of regression, and B) how were outliers identified, and on what basis were they excluded?

This is now re-worded: “We fitted death and bite data from each study to an ordinary least square regression to calculate a case-fatality rate, after removing the extreme outliers. We estimated a crude case-fatality rate of 3.2% for in-hospital cases”.

7) Please also provide a figure corresponding to Figure 1 showing the spatial uncertainty in the estimated risks (Perhaps in the Appendix).

Now a new statistical appendix (Appendix 3) added to our paper. Appendix 3—figure 3 provides the 95% credible interval of absolute risk for each grid cell for Figure 1. The appendix 3—figure 1 shows the distribution of the 7,400 sampling units, demonstrating that the distributed sampling of the MDS would not likely miss the clustering patterns of snakebite deaths (see reply 12).

8) Could you please clarify in the Introduction the purpose of the systematic literature review. It currently states that "We further quantify the levels of envenomations based on…". It appears as though this review adds details on the specific causes, bite locations, and treatment/response to envenomations, while also supporting estimates from the initial analyses.

Corrected. Done.

9) Subsection “Trends in snakebite mortality and its geographic and temporal patterns”, second paragraph: Are the estimates of 55,000 and 61,000 over the same period of time? They are stated as "2001-2" and "2014".

We corrected “2001-2” as “2001”. This was a typo.

10) Please justify the inclusion of "female illiteracy" in the Poisson model.

We have now justified the inclusion of “female illiteracy” as a proxy for socio-economic level in the Materials and methods section, as well as providing the sources of the other variables.

11) There is a lack of general consistency in precision of quantitative estimates reported and a general difficulty in differentiating reported numbers versus estimates. Some numbers are reported in general terms "Approximately" etc., whereas others are reported with greater precision, with corresponding uncertainty. While the use of approximately is sufficient in narrative contexts, there are imprecise numbers that do not have a precise version elsewhere in the manuscript. The mixture of reporting of numbers means it is unclear which numbers are estimates, which are reporting of summary statistics from the million deaths studies, and which are reporting of alternative figures. Clearly differentiating estimates from aggregate reports form the study is critical. Table 2 in particular is confusing, and is lacking any uncertainty reporting. You are recommended to use the GATHER statement https://journals.plos.org/plosmedicine/article?id=10.1371/journal.pmed.1002056 as a way of critiquing the current methods and results summarization, as this provides a good checklist for many of the areas that are lacking in this article.

We have modified Table 2 to include the uncertainty based on diagnosis of snakebite deaths, as this is the major source of uncertainty.

We have also ensured use of one decimal place in most of tables and figures.

12) What is the nature of the representativeness of the Million Deaths Study? This is particularly important for the geospatial analysis since it is not particularly apparent what the spatial distribution of surveyed locations is. A map of locations, or administrative units should be supplied should permissions allow. Even when not, such as some DHS surveys, protocols are reported and summaries presented so that spatial variation can be appreciated. While references are provided in the article, pertinent information should be supplied in the main manuscript, or appendix to flesh this out.

MDS shares the same study sample that official vital statistic generated by SRS for India. Appendix 3—figure 1 provides the map of the MDS sampling areas, showing good national coverage, and addressing the implicit concern by the reviewer about the detection of clusters of snakebite deaths. Because the MDS is so widely distributed, we do not believe that this is a problem. The MDS details and those of the SRS design have been extensively published (see Dhingra et al., 2010, Gomes et al., 2017). Moreover, detailed descriptions of the primary data sources are published online (http://censusindia.gov.in/vital_statistics/SRS_Statistical_Report.html). We prefer not to provide all these details to readers.

13) No references or discussion of the provenance of covariates used in the geospatial model are provided. Where do these numbers come from? What is their spatial resolution? How is uncertainty propagated, where relevant? For the local estimates, what was the source of the corresponding local population estimates? If census statistics, how were these translated to the gridded resolution of Figure 1?

We have now provided the data sources and spatial resolution of the covariates in a new Appendix 3 and in the Materials and methods section. The local population data used in Poisson regression model came from the Sample Registration System (SRS) and they were linked to the mortality data. We have now specified this in the Materials and methods section. Further explanation on how the spatially-smoothed gridded estimates were derived can be found in the newly added Appendix 3 – statistical supplement.

14) What are the implications of potential bias in estimates of using mainly hospital derived cfrs and envenomings where there is likely a more urban bias to rural populations, where the nature of healthcare access and treatment seeking behavior could be different. This could be accentuated as different snake species will differentially impact populations due to differences in life history traits altering how they interact with humans. There is not much discussion as to how (a) these forms of bias may exist in the data and (b) what is the expectation of bias in quantitative estimates. The amenable nature of snakebite mortality means that where the potential for fatality to be averted (in a clinical context) may not be equally available for all participants in the MDS, or rural populations more generally, the literature derived CFR might be of limited external validity.

We thank the reviewer for these points. Indeed, these are useful points that add to the limitations of the systematic review. Moreover, the reviewer rightly points out the varying habits of the snakes involved and different circumstances of the bites differentially affect human populations. For example, anecdotal experience and some published studies indicate that most bites from kraits (*Bungarus* ssp) occur at night while people are sleeping on mats on the floor or ground, in or near home, unprotected by mosquito nets. Most bites from saw-scaled vipers happen either when the snake is stepped upon in bare or sandaled feet at night or when cutting grass by hand with a short sickle.

Bites by cobras are divided into circumstances such as defensive bites while stepping on them during planting/harvesting crops, reaching into piles of straw or firewood and predatory bites when the cobra mistakes a human hand or foot for a prey item. Cobras are also responsible for an unknown percentage of bites on sleeping persons in or near a home.

Russell's viper bites occur at night when a person walks without using a light and steps on one and during the day while harvesting different crops, for example hand harvesting of clumps of leafy peanut plants. Many bites are inflicted on famers in paddy fields or while harvesting other crops. Bites that occur at night are a difficult problem because there is unlikely to be a manned rural health center open. We have added some of the above points to the Discussion as well as some references.

15) The reporting of the geostatistical model is insufficient. It is unclear what the form of the model is and what the equations defining the relationships are. No validation of the model statistics are provided either quantitatively, or in visual reference to a map of snakebite locations. No uncertainty is provided for the estimates. Similar critiques are associated with the use of spline estimates – how was uncertainty propagated here?

See also reply 13. Appendix Section C now provides a map of MDS/SRS sampling areas, a map of MDS snakebite death locations (and crude death rates), and the 95% uncertainty for predicted risks. Appendix 3 provides additional details of the geostatistical model, including model form, formula, validation, and spatial uncertainty of the estimated risks.